# Kin discrimination promotes horizontal gene transfer between unrelated strains in *Bacillus subtilis*

Polonca Stefanic [1,5,6 ✉], Katarina Belcijan[1,5], Barbara Kraigher [1], Rok Kostanjšek[1], Joseph Nesme[2], Jonas Stenløkke Madsen[2], Jasna Kovac [3], Søren Johannes Sørensen [2], Michiel Vos [4] & Ines Mandic-Mulec [1,6 ✉]

*Bacillus subtilis* is a soil bacterium that is competent for natural transformation. Genetically distinct *B. subtilis* swarms form a boundary upon encounter, resulting in killing of one of the strains. This process is mediated by a fast-evolving kin discrimination (KD) system consisting of cellular attack and defence mechanisms. Here, we show that these swarm antagonisms promote transformation-mediated horizontal gene transfer between strains of low relatedness. Gene transfer between interacting non-kin strains is largely unidirectional, from killed cells of the donor strain to surviving cells of the recipient strain. It is associated with activation of a stress response mediated by sigma factor SigW in the donor cells, and induction of competence in the recipient strain. More closely related strains, which in theory would experience more efficient recombination due to increased sequence homology, do not upregulate transformation upon encounter. This result indicates that social interactions can override mechanistic barriers to horizontal gene transfer. We hypothesize that KD-mediated competence in response to the encounter of distinct neighbouring strains could maximize the probability of efficient incorporation of novel alleles and genes that have proved to function in a genomically and ecologically similar context.

[1] Biotechnical Faculty, University of Ljubljana, Ljubljana, Slovenia. [2] Department of Biology, University of Copenhagen, København, Denmark. [3] Department of Food Science, The Pennsylvania State University, University Park, PA, USA. [4] European Centre for Environment and Human Health, University of Exeter Medical School, Environment and Sustainability Institute, Penryn, UK. [5] These authors contributed equally: Polonca Stefanic, Katarina Belcijan. [6] These authors jointly supervised this work: Polonca Stefanic, Ines Mandic-Mulec. ✉email: Polonca.Stefanic@bf.uni-lj.si; Ines.MandicMulec@bf.uni-lj.si

The spore-forming bacterium *Bacillus subtilis* is found in soil- and gut environments and is arguably the best-studied gram-positive model species[1]. *B. subtilis* swarms over surfaces and has diversified into a vast diversity of strains able to recognise non-kin swarms, resulting in the formation of clear swarm boundaries[2]. Kin discrimination (KD) in *B. subtilis* is mediated by a rich arsenal of intercellular attack and defence molecules with extensive variation in transcription levels upon encounter of non-kin[3]. KD genes are present in unique combinations in different strains and likely frequently acquired through horizontal gene transfer[3]. The combinatorial nature of the *B. subtilis* KD system means that genomically divergent strains generally also differ to a greater degree in their carriage of antimicrobial genes. As a result, swarm boundaries between unrelated strains (i.e. non-kin) are very distinct, whereas genomically highly similar strains (i.e. kin) exhibit swarm merging[2].

KD-mediated barriers to swarm-merging result in the territorial sorting of strains according to genetic relatedness during the colonisation of plant roots[2]. However, it is not well-understood whether interference competition is the prime selective force underlying the radiation into many KD types, or whether this mechanism could have other functions. Another explanation for bacterial KD is that it could facilitate horizontal gene transfer between unrelated strains[4]. Recognition and lysis of neighbouring genotypes via the release of effectors by the T6SS secretion system coupled to natural transformation have been demonstrated in the gram-negative species *Vibrio cholerae*[5] and *Acinetobacter baylyi*[6,7]. In the gram-positive species *Streptococcus pneumoniae*, bacteriocin release can result in lysis of neighbouring susceptible genotypes and likewise increase transformation-mediated horizontal gene transfer[8,9].

*B. subtilis* is naturally competent[10,11] but transformation has mostly been studied in the context of single clones growing in liquid medium in this species (but see refs. [12–14]), precluding the action of social interactions such as swarming found in structured environments. We hypothesised that the antagonisms observed between genetically distinct *B. subtilis* strains could lead to transformation-mediated recombination.

Here, we show that swarm boundary interactions between non-kin *B. subtilis* strains result in horizontal gene transfer mediated by the cell-envelope stress-response (*sigW*) in the donor strain, whereas interactions between more closely related kin strains that do not form boundaries do not result in increased recombination. Our results demonstrate that competence regulation mediated by social interactions can be more important than sequence homology for successful recombination.

## Results

**Transformation rates are elevated at swarm boundaries between non-kin strains.** To test whether transformation-mediated gene transfer occurs in swarm boundaries between different *B. subtilis* strains, we first tested for swarm boundary formation between all pairwise combinations of six *B. subtilis* strains isolated from two 1-cm³ soil samples[15] and calculated Average Nucleotide Identity (ANI)[16], which is defined as the mean nucleotide identity of orthologous gene pairs shared between two microbial genomes[17] (Table 1). Strain merging was dependent on genetic relatedness: highly related kin strains (99.93–99.99% ANI) as well as differentially marked isogenic strains ('self' pairings) merged during swarming (merging phenotype), whereas non-kin strains (98.73–98.84% ANI) did not merge and formed clear boundaries (Fig. 1). To test whether transformation-mediated gene transfer occurs more frequently in swarm pairs that form boundaries compared to swarm pairs that merge, differentially marked (Spectinomycin (Sp^R) and Chloramphenicol (Cm^R)) pairs of *B. subtilis* strains (Table 2) were allowed to swarm on an agar surface, after which the swarm meeting area was sampled in order to quantify transformation frequency (the proportion of cells carrying both antibiotic markers as a consequence of transformation-mediated horizontal gene transfer) (Fig. 1a). Transformation frequencies were significantly higher for non-kin strain pairs compared to kin strain pairs or the isogenic controls (two tailed *t*-test; $p < 0.05$) (Fig. 1b and Supplementary Fig. 1a, b). Experiments using strains with inverse marker combinations yielded statistically identical results (Supplementary Figs. 1–2 and Supplementary Notes 1). Mixing of differentially marked strains (1:1) in shaken co-cultures did not result in increased transformation of non-kin DNA (Supplementary Fig. 1c, d), indicating that surface-based cell contact is required for KD-mediated transformation.

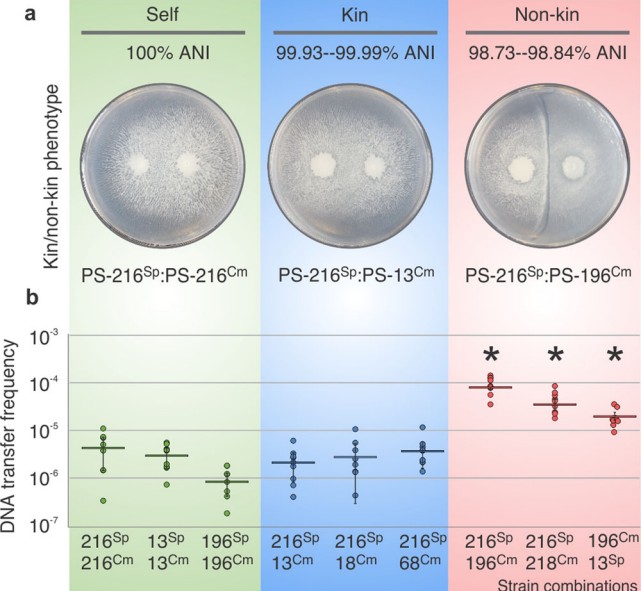

**Fig. 1 Transformation frequency as a function of genomic relatedness between strains. a** Representative interactions of control self-interaction (green), kin interaction (blue) and non-kin interaction (red) (ANI indicated). **b** Transformation frequency as quantified by the frequency of double mutants isolated from the swarm boundary ($n = 3$ biologically independent experiments, each $n$ performed in three technical replicates (9 data points), error bars represent SD). Asterisks represent statistically significant values compared to corresponding self and kin pairings (two tailed Student's *t*-test for unpaired data assuming equal variances), (Supplementary Notes and Supplementary Fig. 2). Data are presented as mean values ± SD and error bars represent SD of the mean values ($n = 3$). For strain abbreviations see Table 2.

**Table 1 Average Nucleotide Identity (ANI) of strain pairs.**

|        | PS-13  | PS-18  | PS-216 | PS-68  | PS-196 | PS-218 |
|--------|--------|--------|--------|--------|--------|--------|
| PS-13  | 1.0000 |        |        |        |        |        |
| PS-18  | 0.9999 | 1.0000 |        |        |        |        |
| PS-216 | 0.9997 | 0.9994 | 1.0000 |        |        |        |
| PS-68  | 0.9996 | 0.9993 | 0.9995 | 1.0000 |        |        |
| PS-196 | 0.9882 | 0.9881 | 0.9883 | 0.9884 | 1.0000 |        |
| PS-218 | 0.9873 | 0.9875 | 0.9875 | 0.9875 | 0.9877 | 1·0000 |

Pairwise Average Nucleotide Identity (ANI) is shown for *B. subtilis* strain pairs.

**Transformation is largely unidirectional, with one swarm principally acting as a recipient and the other as a donor**. The previous experiment (Fig. 1) was based on the detection of double recombinants, which in theory could arise from successful DNA uptake and recombination by both strains or by only one strain. To test the directionality of DNA transfer, we constructed *comGA* mutants with impaired DNA uptake efficiency[18,19]. Four wild-type strains were paired both with a *comGA* self- control and with a non-kin *comGA* mutant and double transformants at the swarm boundary between strains were enumerated. The transformation frequencies were below the level of detection for some wild-type strains (zero transformants were obtained), especially in self and kin setting, but transformation frequency generally increased above the detection limit in non-kin pairings (Fig. 2). When PS-

216 was paired with a competence defective PS-196 Δ*comGA*, its transformation frequency increased significantly (~7-fold) relative to the isogenic control (two tailed *t*-test, $p = 0.0002$). PS-216 also showed an increased transformation frequency when paired with PS-218 Δ*comGA*, however not significantly (two tailed *t*-test, $p = 0.1161$). Similarly, PS-196 and PS-13 displayed increased transformation frequencies when paired with the PS-13 and PS-196 Δ*comGA* strains, respectively, even though the transformation frequencies of self- controls were below the detection limit (Fig. 2). In the reversed examples above, where the PS-196 and the PS-218 strains were paired with PS-216 Δ*comGA*, no recombination could be detected in the recipient strains PS-196 and PS-218, respectively. No transformants could be detected when pairing two differently marked Δ*comGA* mutants (data not shown), confirming that horizontal gene transfer during swarming is mediated by the activation of competence. This experiment demonstrates that gene flow is primarily unidirectional.

**Swarm boundaries are a site of asymmetric cell lysis**. Scanning electron microscopy revealed that areas where kin strains merged contained healthy intact cells, whereas swarm boundaries between non-kin strains showed deflated and empty cells, pointing to cell lysis (Fig. 3a). To test whether cell lysis occurred predominantly in the strain serving as the DNA donor for transformation (Fig. 2), we first compared the fitness of strains (i.e. cell survival–$CFU_{strainA}/CFU_{strainA+B}$) during non-kin vs self-interaction and the fitness of strains in kin vs self- interaction. Fitness of strains in kin setting was not significantly different from the fitness of self-paired strains (isogenic marked mutants) (PS-216:PS-13, $p_{216} = 0.58$, $p_{13} = 0.55$; PS-216:PS-18, $p_{216} = 0.92$, $p_{18} = 0.94$; PS-216:PS-68, $p_{216} = 0.95$, $p_{68} = 0.53$). However, in non-kin strain pairs, fitness of interacting strains was significantly different from that during self-interaction (PS-216:PS-196, $p_{216} = 0.01$, $p_{196} = 0.01$; PS-216:PS-218, $p_{216} = 0.003$, $p_{218} = 0.03$; PS-13:PS-196, $p_{13} = 0.004$, $p_{196} = 0.005$).

| Table 2 Strain abbreviations. | | |
|---|---|---|
| **Strain name** | **Strain genotype** | **Strain abbreviation** |
| PS-216 | Wt | 216 |
| | p43-*cfp* (Sp) | 216$^{Sp}$ |
| | p43-*yfp* (Cm) | 216$^{Cm}$ |
| PS-18 | Wt | 18 |
| | p43-*cfp* (Sp) | 18$^{Sp}$ |
| | p43-*yfp* (Cm) | 18$^{Cm}$ |
| PS-68 | Wt | 68 |
| | p43-*cfp* (Sp) | 68$^{Sp}$ |
| | p43-*yfp* (Cm) | 68$^{Cm}$ |
| PS-13 | Wt | 13 |
| | p43-*cfp* (Sp) | 13$^{Sp}$ |
| | p43-*yfp* (Cm) | 13$^{Cm}$ |
| PS-196 | Wt | 196 |
| | p43-*cfp* (Sp) | 196$^{Sp}$ |
| | p43-*yfp* (Cm) | 196$^{Cm}$ |
| PS-218 | Wt | 218 |
| | p43-*cfp* (Sp) | 218$^{Sp}$ |
| | p43-*yfp* (Cm) | 218$^{Cm}$ |

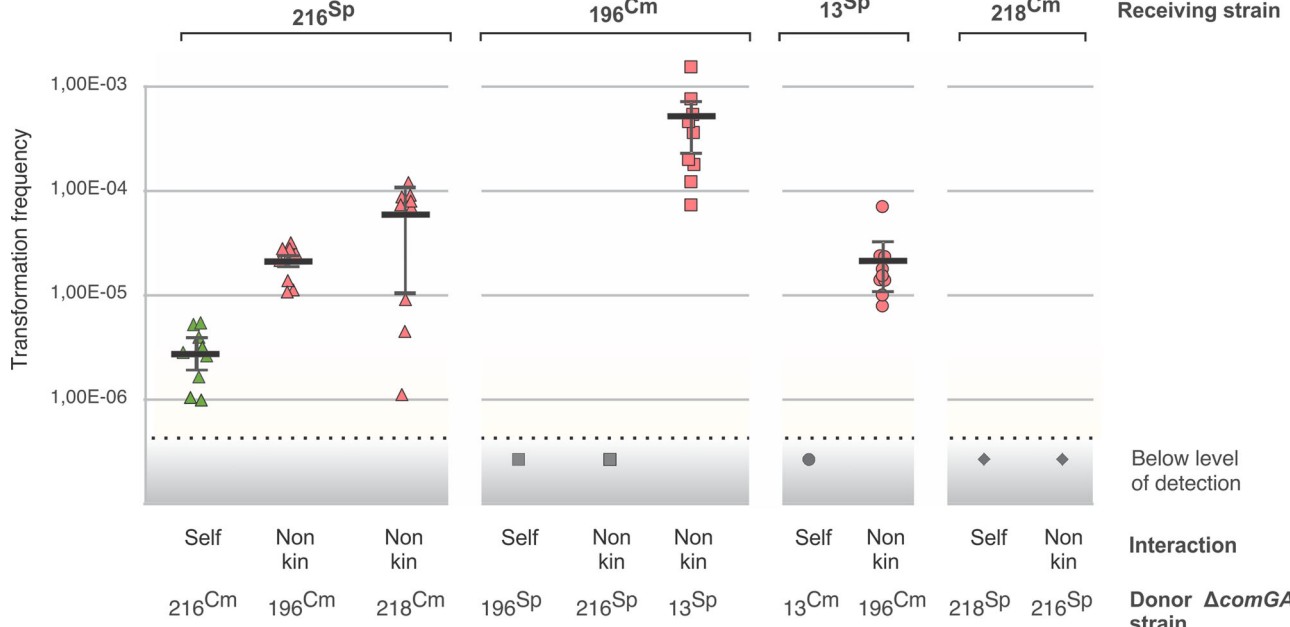

**Fig. 2 Transformation frequencies of wt *B. subtilis* DNA receiving strains paired with donor Δ*comGA* mutants.** Transformation frequency was measured as the proportion of double transformants at swarm boundaries. Self-interactions are shown in green, non-kin interactions in red and transformation frequencies below level of detection in grey colour. Data are presented as mean values ± SD and error bars represent SD of the mean values ($n = 3$ biologically independent experiments, each *n* performed in three technical replicates (9 data points). Strains are depicted by different shapes: PS-216 triangle, PS-196 square, PS-13 circle and PS-218 a diamond. For strain abbreviations see Table 2 and for strain combinations see Supplementary Table 3.

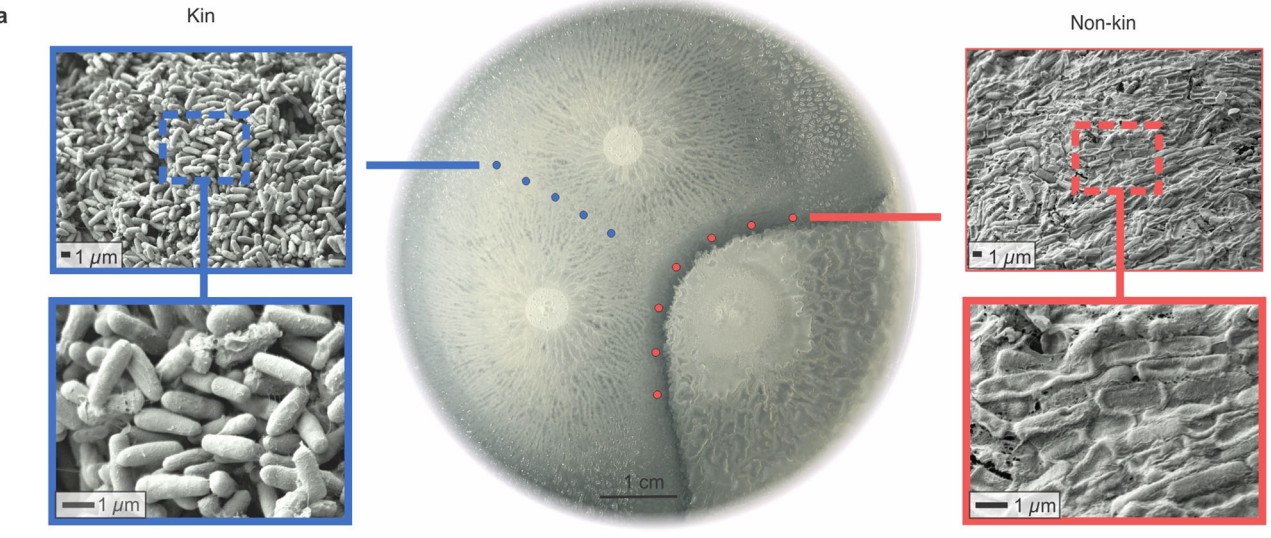

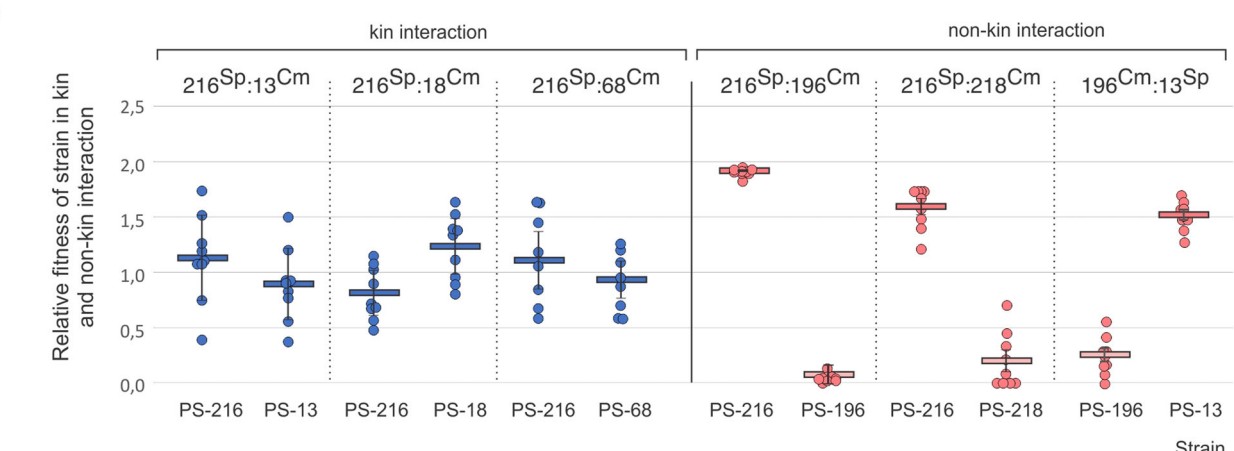

**Fig. 3 Kin discrimination mediates fitness via cell lysis. a** Swarm assay on B agar. The left and right panels are scanning electron microscopy micrographs taken at the meeting point of kin strains PS-216$^{Sp}$ and PS-13$^{Cm}$ (blue, left) and non-kin strains PS-216$^{Sp}$ and PS-196$^{Cm}$ (red, right). **b** Relative fitness of strains in kin pairings (blue) and non-kin pairings (red) ($n = 3$ biologically independent experiments, each $n$ performed in three technical replicates (9 data points)). See Table 2 for strain abbreviations. All experiments (in **a** and **b**) were performed in three biologically independent experiments. Data are presented as mean values ±SD and error bars represent SD of the mean values ($n = 3$).

Furthermore, we compared the relative fitness of three non-kin strain pairs at the swarm interface to the relative fitness of each strain staged with a kin strain (Fig. 3b). For example, relative fitness of PS-216 in non-kin setting was determined by calculating fitness of strain PS-216$^{Sp}$ when paired with a non-kin PS-196$^{Cm}$ strain and divided by the fitness of the PS-216$^{Sp}$ strain when paired with a differentially marked version of itself (PS-216$^{Cm}$), see Eq. (1). Similarly, relative fitness of strain PS-216$^{Sp}$ in a kin setting was calculated as fitness of the PS-216$^{Sp}$ strain when paired with PS-13$^{Cm}$ and divided by the fitness of the PS-216$^{Sp}$ when paired with differentially marked self-strain PS-216$^{Cm}$. We observed an increase in the relative fitness of one strain and a decrease of relative fitness in the other strain in all three non-kin interactions (Fig. 3b). This is consistent with the scanning electron microscope observations (Fig. 3a) and suggests lysis of one strain by the other strain in non-kin encounters: the PS-216 strain kills the PS-196 and PS-218 strain, and PS-13 kills the PS-196 strain.

**DNA release through cell lysis is not required for efficient transformation.** As transformation is dependent on the presence of extracellular DNA, we next sought to test whether transformation in swarm boundaries is facilitated because cell lysis increases DNA availability. We first confirmed that extracellular DNA is a prerequisite for DNA uptake by measuring transformation on agar plates supplemented with DNases which degrade free DNA (Supplementary Fig. 3a). Cells were harvested from swarm boundaries sections of the agar surface that were either treated with DNaseI or were left untreated in order to quantify double recombinants (Supplementary Fig. 3a). Numerous transformants were obtained from non-DNase-treated agar sections, but no transformants could be recovered from the DNase treated portion of the agar plate, confirming that extracellular DNA is a prerequisite for the observed horizontal gene transfer events (Supplementary Fig. 3b).

Next, we tested whether cell lysis might increase extracellular DNA concentrations at the boundary and promote DNA transfer with non-kin. Surprisingly, extracellular DNA concentrations measured at the boundary between non-kin strains were comparable to those between staged kin- or isogenic strains (Fig. 4a and Supplementary Fig. 4). To test for the possibility that extracellular DNA is degraded by DNases to the extent markers

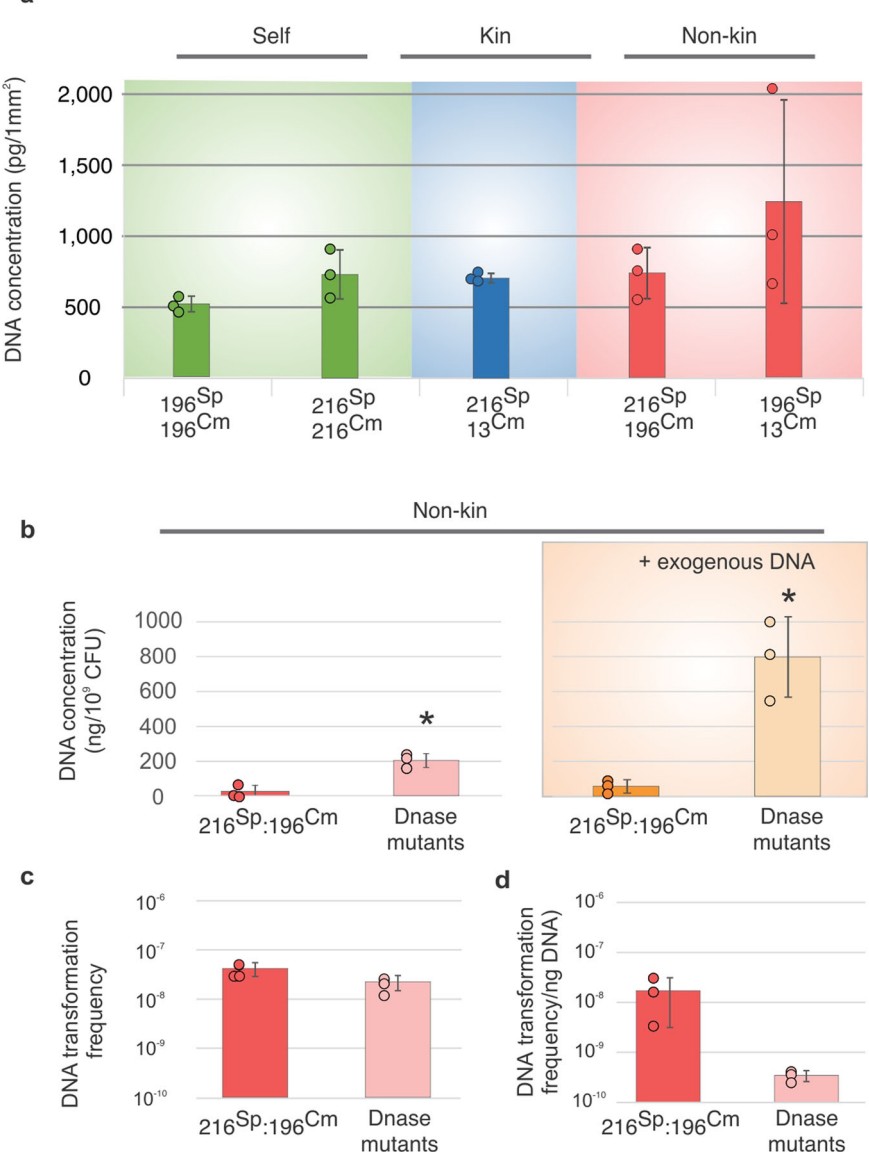

**Fig. 4 Transformation frequency at swarm boundaries is largely unaffected by DNA concentration. a** DNA concentrations at the meeting points of self-controls (green), kin (blue) and non-kin strains (red), **b** DNA concentration at the swarm boundary between non-kin wt PS-216<sup>Sp</sup> and PS-196<sup>Cm</sup> (red) and between nuclease mutants (DNase mutants) PS-216<sup>Sp</sup> ΔyhcR ΔnucB and PS-196<sup>Cm</sup> ΔyhcR without exogenously added DNA (pink) (left panel) and with 30 μg DNA added at the swarm boundary (right panel, wt orange, DNase mutants yellow), asterisks represent statistically significant values compared to corresponding wt pairings (two tailed Student's *t*-test for unpaired data assuming equal variances), **c** transformation frequency at the boundary of two non-kin wt (PS-216<sup>Sp</sup> and PS-196<sup>Cm</sup>) (red) and between two nuclease mutant strains (PS-216<sup>Sp</sup> ΔyhcR ΔnucB and PS-196<sup>Cm</sup> ΔyhcR) (pink), **d** transformation frequency at the boundary of two non-kin wt strains (PS-216<sup>Sp</sup> and PS-196 <sup>m</sup>) (red) and between two nuclease mutants strains (PS-216<sup>Sp</sup> ΔyhcR ΔnucB and PS-196<sup>Cm</sup> ΔyhcR) (pink) calculated per ng available DNA. Assays were replicated three times. See Table 2 for strain abbreviations. Data are presented as mean values ±SD and error bars represent SD of the mean values (n = 3). For statistical parameters see Supplementary Notes.

cannot be successfully taken up and recombined, we sampled the swarm meeting area and extracted DNA from the agar samples following a spectrophotometric DNA quantification (Supplementary Notes 2). We measured DNA concentrations at the boundary of two wt strains (PS-216 and PS-196) and compared it to the DNA concentration measured between two nuclease mutant strains (PS-216<sup>Sp</sup> ΔyhcR, ΔnucB and PS-196<sup>Cm</sup> ΔyhcR; see Supplementary Table 3). Extracellular DNA concentrations were significantly higher at the boundary between non-kin nuclease mutant strains compared to wt strains (two tailed *t*-test, $p = 0.004$). The activity of extracellular nucleases was furthermore demonstrated in another experiment where 30 μg of DNA extracted from PS-216 ΔepsA-O (Tet<sup>R</sup>) was added in the middle

of agar plates inoculated with non-kin strains. Significantly lower DNA concentrations were measured on the boundary between two wt non-kin strains, whereas the non-kin nuclease mutants showed elevated DNA concentrations at the boundary ($p_{+DNA} = 0.005$) (Fig. 4b and Supplementary Fig. 4). However, elevated concentrations of extracellular DNA did not significantly affect transformation frequency between two non-kin nuclease mutant strains. In fact, transformation frequency between nuclease mutants was not significantly different from corresponding wild-type strains in the presence of added DNA (two tailed *t*-test, $p = 0.093$) (Fig. 4c and Supplementary Fig. 4b). These combined results indicate that the concentrations of extracellular DNA in swarms are saturating (i.e. not limiting transformation)

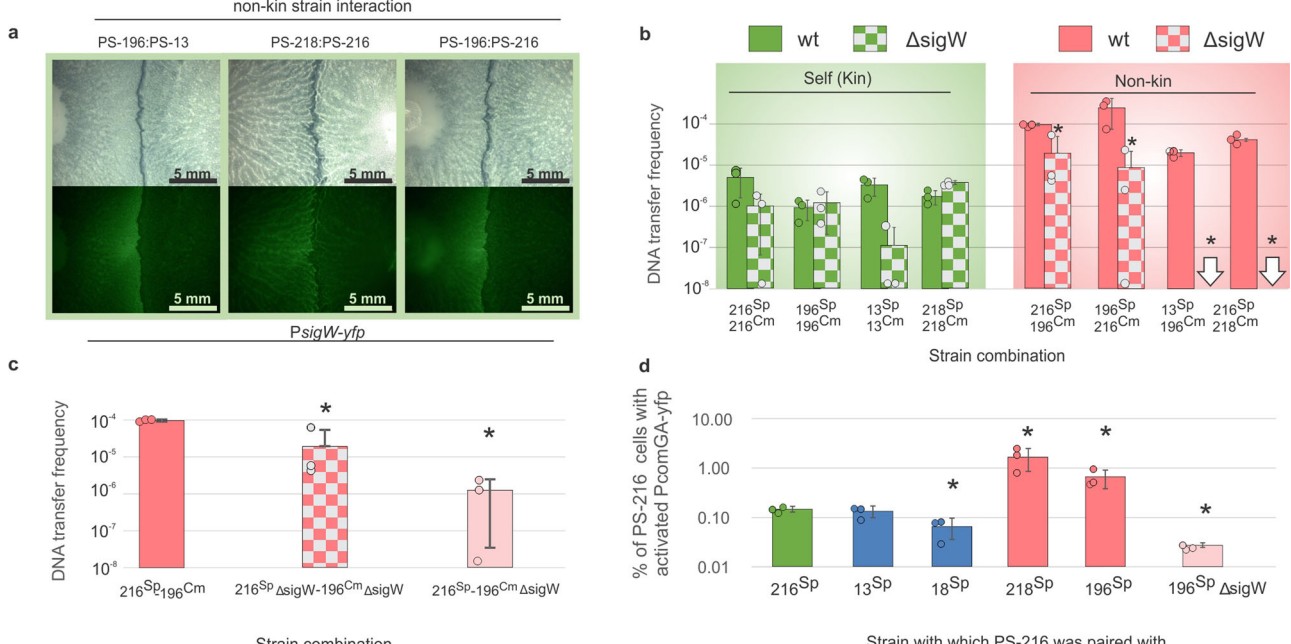

**Fig. 5 Cell envelope stress and competence induction in swarm boundaries. a** Meeting points of non-kin *sigW-yfp* swarms. **b** The effect of stress response inactivation (Δ*sigW*) on transformation rate. Green depicts control self-combinations and red depicts non-kin combinations; full columns and chequered columns represent wt controls and Δ*sigW* mutants respectively. Asterisks represent statistically significant values (two tailed Student's *t*-test for unpaired data assuming equal variances, $p_{216-196} = 0.01$, $p_{196-216} = 0.05$). **c** The effect of *sigW* inactivation in a non-kin donor strain on transformation frequency in recipient strain PS-216. Red column = PS-216 and PS-196 wt control donor, chequered column = PS-216 Δ*sigW* and PS-196 Δ*sigW* donor, pink column = PS-216 and PS-196 Δ*sigW* donor. Asterisks represent statistically significant values (two tailed Student's *t*-test for unpaired data assuming equal variances, $p_{WT-216\Delta sigW:196\Delta sigW} = 0.01$; $p_{WT-216:196\Delta sigW} = 0.00002$). **d** Activation of competence genes in strain PS-216 *comGA-yfp* at the meeting point with self (PS-216$^{Sp}$)(green), kin strains (PS-13, PS-18) (blue) and non-kin strains (PS-196, PS-218 (red) and Δ*sigW* PS-196 (pink)). Bars represent fractions of PS-216 cells expressing the *comGA* gene at swarm meeting points. Data are presented as mean values ± SD and error bars represent SD of the mean values. Assays were replicated three times. Asterisks represent statistically significant values (two tailed Student's *t*-test for unpaired data assuming equal variances, $p_{216-18} = 0.02$, $p_{216-218} = 0.03$, $p_{216-196} = 0.03$, $p_{216-196\Delta sigW} = 0.0005$, $p_{196-196\Delta sigW} = 0.02$, see Supplementary Notes). See Table 2 for strain abbreviations and for strain combinations see Supplementary Table 3.

and that cell lysis in swarm boundaries is not a prerequisite for efficient recombination (Fig. 4 and Supplementary Notes 2).

**Increased transformation rate in swarm boundaries is a consequence of cell envelope stress-induced upregulation of competence.** Stress and the release of antimicrobials have been shown to induce competence in a variety of bacterial species[20–23]. In *B. subtilis*, the *sigW* ($\sigma^W$) gene is expressed in response to cell envelope stress[24] and was previously shown to be upregulated at the non-kin swarm boundary[3]. To examine the connection between stress and competence, we visualised the expression of P*sigW-yfp* (σW) and measured transformation rates in Δ*sigW* ($\sigma^W$) mutants, both in non-kin swarm boundaries and in merging areas between kin strains. First, we tagged strains PS-13, PS-196, PS-216 and PS-218 with P*sigW-yfp* and observed an increased YFP signal in non-kin swarm boundaries, whereas kin and self-interactions showed no P*sigW-yfp* induction (Supplementary Fig. 5). *sigW* was predominantly induced in the strain that was antagonised (Fig. 5a and Supplementary Fig. 5), corresponding with the results of the pairwise fitness assays (Fig. 3b).

Since stress could induce competence, we next tested whether an inability to respond to stress (Δ*sigW*) prevents the induction of competence and lowers transformation frequency. Indeed, when non-kin PS-216$^{Sp}$Δ*sigW* and PS-196$^{Cm}$Δ*sigW* were staged, transformation rates decreased significantly compared to the wt control pairing (two tailed *t*-tests, $p = 0.01$). The decreased transformation rates of Δ*sigW* mutants were statistically indistinguishable to transformation frequencies between wt self- strain

pairs (Fig. 5b) (Supplementary Notes 3). Similarly, when we compared two other non-kin Δ*sigW* strain pairs (PS-13:PS-196 and PS-216:PS-218), transformation frequencies decreased dramatically compared to wt strains as no transformants were detected in non-kin Δ*sigW* strain pairs (Fig. 5b). No significant changes occurred to the rate of transformation in isogenic combinations, apart from PS-13, where Δ*sigW* strains pairs showed reduced transformation (Fig. 5b and see Supplementary Notes 3).

How can the fact that the dominant strain acts as the recipient of DNA (Fig. 2) be reconciled with the finding that DNA uptake and recombination is associated with a stress response (*sigW* expression) which can be expected to be most prominent in the lysed (not the lysing) strain (Fig. 2)? We hypothesised that the *sigW* expression in the lysed strain induces competence in the lysing strain. To test this, we staged the wt PS-216 strain with a PS-196 Δ*sigW* mutant (the latter strain undergoes lysis in this pairing), (Fig. 3b) and measured transformation rates). Consistent with our hypothesis, wt PS-216 transformation frequency was significantly lower in this strain combination compared to its transformation frequency when staged with the PS-196 wt strain ($p = 0.00002$) (Fig. 5b). Transformation frequency of PS-216 paired with PS-196 Δ*sigW* mutant was statistically indistinguishable to transformation frequency between strains both carrying the Δ*sigW* mutation ($p = 0.288$) (Fig. 5c). This indicates that *sigW* induction in the lysed strain is the cause of increased transformation rate in the strain causing lysis.

To verify whether higher transformation frequencies during non-kin encounters are due to the upregulation of competence,

we tagged the central competence gene *comGA* in strain PS-216 with a YFP reporter and paired this strain with: (i) a wt version of itself (PS-216), (ii) two wt kin strains (PS-13 and PS-18) and (iii) two wt non-kin strain (PS-196 and PS-218) (Fig. 5d). Significant upregulation of *comGA* in PS-216 was observed at the boundary with non-kin strains, compared to *comGA-yfp* activity of self (two tailed *t*-test, PS-216:PS-196, $p = 0.031$; PS-216:PS-218, $p = 0.032$). *comGA* expression of PS-216 in kin and self- encounters was either statistically the same (two tailed *t*-test, PS-216:PS-13, $p = 0.598$) or even lower (two tailed *t*-test, PS-216:PS-18, $p = 0.017$) (Fig. 5d). To further test the link between stress response and competence upregulation, we paired the PS-216 strain carrying the *comGA-yfp* with a PS-196 Δ*sigW* and observed down regulation of *comGA* (Fig. 5d). These experiments demonstrate that in non-kin swarm boundaries competence in the recipient strain is induced by cell envelope stress in the antagonised donor strain.

**KD-mediated transformation has the potential to speed up adaptation**. The evolutionary benefits of competence and transformation are still the subject of active debate and several, non-mutually exclusive benefits have been proposed including nutrient acquisition, repair and sex-like benefits[25,26]. As the uptake and recombination of extracellular DNA is an integral part of transformation, it is often assumed that recombination-based benefits are at least partly responsible for its evolutionary maintenance[25]. A proof-of-principle experiment was devised to test whether KD-mediated transformation could aid adaptation through the uptake of beneficial alleles originating from the encounter with a non-kin swarm. Agar plates were prepared such that only one half contained Sp and Cm antibiotics (Fig. 6a). Pairs of either kin- or non-kin strains differentially marked with Sp and Cm were inoculated on the half of the agar plate unamended with antibiotic and allowed to swarm (Fig. 6a). Staged kin strain pairs mainly stayed on the half of the plate unamended with antibiotics, whereas non-kin combinations were able to invade the antibiotic-amended portion in the plate (Fig. 6b and Supplementary Fig. 6). In all, 100% and 98% of cells sampled from two non-kin pairing-derived invading swarms were resistant to both antibiotics and carried double fluorescence markers CFP and YFP (Supplementary Fig. 7 and Supplementary Notes 4). These findings confirm that invading lineages were the result of transformation-mediated recombination events between non-kin strains. A control experiment where paired non-kin strains did not carry Sp and Cm resistance genes, did not lead to swarm invasions (Supplementary Fig. 8).

## Discussion

Here we demonstrate that kin discrimination promotes horizontal gene transfer in *B. subtilis* through upregulation of competence in response to cell envelope stress. *B. subtilis* antimicrobial genes (*sunA*, *sdpC*, *sboA*, *yobL*, *skf*, *wapA*, *srfA*) have been shown to be upregulated at the non-kin boundary[3]. It is possible that antibiotics that affect cell-envelope integrity induce *sigW* at the boundary which, together with competence development, increases tolerance to antimicrobials and augment the more competent strain[27]. It could also be speculated that *sigW* activated cells with damaged membranes release stress-related metabolites that upregulate competence in the dominant strain as was documented during inter-species transformation experiments where lysis of *E. coli* cells and subsequent release of cell metabolites affected the transformation ability of *B. subtilis* recipient cells[28]. Similarly, it was recently shown that exposure to antibiotics increases cell-to-cell natural transformation in *B. subtilis*

by affecting the donor strain through as yet unknown mechanism[29].

Induction of competence in *B. subtilis* requires activation of a quorum sensing (QS) system, encoded by the *comQXPA* operon[30,31], that ensures activation of the competence genes via the induction of ComS[32,33]. We and others have shown that *B. subtilis* genotypes evolved extensive polymorphism in the ComQXPA QS system[15,34–36]. In liquid competence media, competence can only be induced if two strains express the same pheromone (i.e. both strains belong to the same phenotype)[15,34,37]. Limited cross-talk and sometimes even inhibition of competence has been observed between different phenotypes[34], leading to the expectation that transformation between phenotypes is lower than within phenotypes. In contrast, the non-kin strains that engage in transformation in this study all belong to different phenotypes[2,15], for example, PS-216 and other kin strains belong to "phenotype 168", whereas PS-218 and PS-196 belong to "NAF4 phenotype"[15], yet still show higher transformation efficiency when paired than do more closely related kin strain pairs belonging to the same phenotype. This discrepancy with earlier studies could potentially result from the fact that different mechanisms could be important governing HGT in structured (agar) environments versus unstructured (broth) environments.

Interestingly, we found the rate of horizontal gene transfer to increase with genomic divergence. This result seems counterintuitive at first, as the efficiency of transformation-mediated recombination of more divergent DNA fragments has been shown to decrease log-linearly with increasing sequence dissimilarity in *Bacillus*[38–40]. Social interactions leading to competence development thus can play a more significant role in the efficiency of uptake of foreign DNA than constraints imposed by the recombination machinery in this species. This highlights the importance of considering social interactions in the study of bacteria generally, and in the study of horizontal gene transfer specifically. A variety of (non-mutually exclusive) explanations for the evolutionary benefits of natural transformation have been put forward, one of which is that increased genetic variation through recombination with foreign DNA facilitates adaptation[25,26] or that it can aid curing the genome of selfish deleterious elements[41]. The observed increase in recombination between genomically more distant strains is not predicted by the DNA for food hypothesis and poses a problem for the DNA repair hypothesis but would be consistent with the sex hypothesis. The coupling of competence to strain-specific killing could ensure that recombination is not upregulated in (near) clonal swarms (which would not introduce any genetic variation) nor upon encounter of dissimilar species (which would unlikely result in successful recombination). Instead, lysis of distinct but closely related strains occupying the same patch maximises the probability of efficient incorporation of novel alleles and genes that have proved to function in a similar genomic and ecological context[42]. Such non-kin interactions are common in natural populations of *B. subtilis*: 84% of strain combinations isolated from two microscale soil aggregates were shown to be non-kin[2]. Evolution experiments incorporating ecological realism are needed to shed light on the adaptive benefits of KD-mediated antagonism and transformation.

## Methods

**Strains and media**. Strains used in this study and their mutant derivatives are presented in Supplementary Table 3. Strain combinations used in experiments are shown in Supplementary Table 3. Briefly, 6 *Bacillus subtilis* wild-type strains isolated from sandy bank of the Sava River in Slovenia[11] were used (Table 2). For strains carrying antibiotic markers appropriate antibiotics at the following concentrations were used: 5 µg/ml of Chloramphenicol (Cm), 100 µg/ml of Spectinomycin (Sp), 20 µg/ml of Tetracycline (Tet), 25 µg/ml of Kanamycin (Kn) and 20 µg/ml of

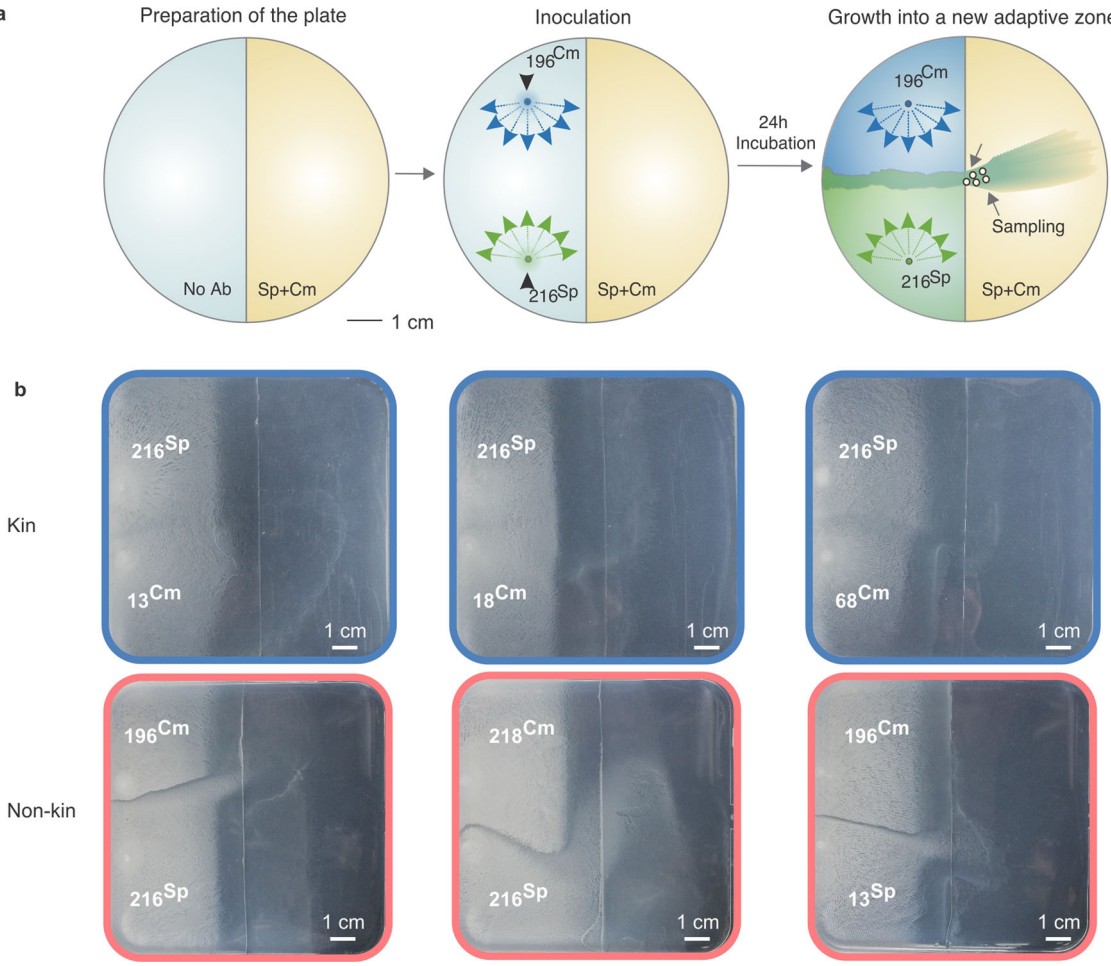

**Fig. 6 Transformation at swarm boundaries can result in recombinants with novel adaptive capabilities. a** Experimental design. Samples were taken from the swarm that spread onto the part of plate containing both antibiotics (Ab): Chloramphenicol (Cm) and Spectinomycin (Sp) (marked with arrows and with circles). **b** Meeting points of kin (blue) and non-kin (red) strain pairs after 24 h of incubation on 12 cm B media agar plates. See Table 2 for strain abbreviations.

Erythromycin (Ery). Overnight cultures were prepared in liquid LB medium and swarming assays were performed on swarming agar (final agar concentration 0.7%), which was based on B-medium composed of 15 mM $(NH_4)_2SO_4$, 8 mM $MgSO_4 \times 7H_2O$, 27 mM KCl, 7 mM sodium citrate $\times 2H_2O$, 50 mM Tris-HCl, pH 7.5; 2 mM $CaCl_2 \times 2H_2O$, 1 μM $FeSO_4 \times 7H_2O$, 10 μM $MnSO_4 \times 4H_2O$, 0.6 mM $KH_2PO_4$, 4.5 mM sodium glutamate, 0.86 mM lysine, 0.78 mM tryptophan and 0.2% glucose.

**Strain construction**. For fluorescence visualisation experiments wild-type *B. subtilis* strains were tagged with a *yfp* or a *cfp* gene linked to a constitutive promoter (p43), inserted at the *sacA* or *amyE* locus, and for observation of competence induction a *yfp* gene was linked to the *comGA* promoter. P43-*yfp* construct from Pkm3-p43-*yfp* plasmid[2] was digested with EcoRI and BamHI and ligated into the ECE174[43] plasmid yielding the pEM1071 plasmid carrying a p43-*yfp* fusion inside the *sacA* integration site. For construction of p43-*cfp* fusion PkM8 plasmid was used, carrying the *cfp* gene linked to *spoIIQ* region[44]. *spoIIQ* region from the original plasmid was removed by EcoRI and HindIII digestion and replaced by p43 promoter sequence, which was amplified with primers p43-F1-EcoRI and p43-R1-HindIII[2] (Supplementary table 2) and was ligated into the PkM8 EcoRI and HindIII sites yielding the Pkm8-p43-*cfp* pEM1069 plasmid.

Strains carrying P*comGA-yfp* (Cm) and knockouts in Δ*nucB* (kn), Δ*yhcR* (Ery), Δ*comGA* (Ery), Δ*epsA-O* (Tet) and Δ*comQXP* (Kn), *sigW-yfp* (Ery) and Δ*sigW* were obtained by transforming the wt *B. subtilis* strains with DNA isolated from *B. subtilis* 11A79 (P*comGA-yfp*)[44], BD2121 (Δ*comK*)[45], BKK25750 (Δ*nucB*)[46], BKE09190 (Δ*yhcR*)[46], BKK24730 (Δ*comGA*, kn)[46], BKE24730 (Δ*comGA*)[46], ZK4300 (Δ*epsA-O*)[3], plasmid pED302 (Δ*comQXP*)[37], NL362 (Δ*sigW*)[3] and ZK4860 (*sigW-yfp*)[3] (Supplementary Table 3). The DNA (plasmid or genomic) was introduced to the *B. subtilis* using standard transformation protocol.

**Harvesting spores**. For swarming assay spores were inoculated on semi solid media and spore crops were prepared by inoculating an overnight *B. subtilis* culture from LB medium to 2 x SG sporulation medium[47]. Cultures were incubated with

shaking (200 rpm) at 37 °C for 3 days, after which spores were washed twice with saline solution and stored in glycerol at −20° degrees.

**Frequency of DNA exchange between *B. subtilis* strains in liquid CM media**. To determine the frequency of DNA exchange between kin or non-kin *B. subtilis* strains, strain pairs were inoculated into a liquid competence medium (CM). CM medium was inoculated with spores (final concentration 1%), where spores of each strain represented half of the inoculum. Inoculated media was shaken (200 rpm) at 37 °C and was sampled after 8 h. Using appropriate antibiotic selection, CFUs of each strain, as well as the number of cells that have taken up each other's DNA, were determined. DNA exchange frequency was calculated as the number of total CFU/ml of transformants (Cm + Sp) divided by total CFU count (sum of CFUs of both strains (Cm and Sp) minus the number of transformants (Cm + Sp)). The frequency of DNA exchange in liquid media was performed in three biologically independent experiments each in three technical replicates for every strain combination.

**Swarm boundary assay**. Swarming assays were performed as previously described[2], with a few minor adaptations. *B. subtilis* spores were diluted 1:1 with LB media and 1–2 μl was inoculated in parallel on semisolid 0.7% agar B media. Pairs of inoculated kin or non-kin strains were labelled with different antibiotic resistance and fluorescence marker. When spores were unavailable, overnight cultures, prepared from the freezing stocks (−80°) were transferred to fresh liquid LB media (1% inoculum) and incubated for 2 h with shaking (200 rpm) at 37 °C. After 2 h incubation, the culture was again transferred to fresh liquid LB media (1% inoculum) and incubated with shaking (200 rpm) at 37 °C for additional 2 h. After the second incubation the exponential phase cell suspension was used for B media inoculation (for experimental scheme see Supplementary Fig. 9). Samples were taken in the same manner for CFU or DNA quantification). Inoculated agar plates were incubated overnight (22–24 h) at 37 °C and 80% humidity. If not stated

otherwise, experiments were performed in three biological replicates and each replicate was sampled in three technical replicates.

**Frequency of DNA exchange between swarms of *B. subtilis* strains on semi-solid B media**. To determine the frequency of DNA exchange between kin or non-kin *B. subtilis* swarms, pairs of strains were inoculated parallel to each other on semisolid B medium and incubated overnight as described above. Ten samples (per plate) were taken at the swarm meeting point with cut pipette tips and resuspended in 500 μl 0.9% NaCl solution. Using appropriate antibiotic selection, CFUs of each approaching swarms at the meeting point was determined as well as the number of cells that have taken up each other's DNA. DNA exchange frequency at the border was calculated as the number of total CFU/ml of transformants (Cm + Sp) divided by total CFU count (sum of CFUs of both strains (Cm and Sp) minus the number of transformants (Cm + Sp)). Experiment was performed in three biological replicates and each replicate was sampled in three technical replicates.

**DNA exchange in the presence/absence of DNaseI**. In order to quantify DNA exchange in the presence of DNaseI, 20 μl of DNaseI (5 mg/ml) was spread on one half of the semisolid B media (concentration on the surface of media was 50 μg/ml) (Supplementary Fig. 3). *B. subtilis* strains were inoculated parallel to each other and incubated overnight for swarms to spread as described above. Sampling was performed at the swarm meeting point of non-kin strains in the area containing DNaseI and in the area without DNaseI present (Supplementary Fig. 3). Ten samples were picked from each area (with/without DNaseI) using cut pipette tips and resuspended in 500 μl 0.9% NaCl solution. CFUs were determined and DNA exchange was calculated as described above. Experiment was performed in three biological replicates.

***comGA* gene expression**. The *comGA-yfp* gene activation was observed at meeting points of kin and non-kin swarms. Pairs of kin and non-kin strains were inoculated parallel to each other on semisolid B media and incubated overnight as described above. Samples were taken at the meeting points of two swarms with an inoculation loop, resuspended in 200 μl of 0.9% NaCl solution and mixed with a vortex mixer. In total, 20 μl of each sample was pipetted onto a glass slide coated with a solution of poly-L-lysine (0.1% (w/v)). Before microscopic observation, slides were dried and antifade reagent SlowFade (Invitrogen, ZDA) was added. Zeiss Axio Observer Z1 with DIC technique (differential interference contrast) and fluorescence technique with fluorescence filter (Zeiss, Göttingen, Nemčija) was used for observing fluorescent protein *yfp* (Ex 430 nm/Em 474 nm). For every technical replicate 20 fields of view were analysed and photographed totalling to ~30,000 cells per studied interaction (self, kin, non-kin and Δ*sigW*). Pictures were analysed with Fiji (1.51d). Overall number of cells (DIC technique) and number of cells expressing *comGA-yfp* (yellow glowing cells under microscope with fluorescence filter for observing *yfp*) were determined. The ratio of cells expressing *comGA-yfp* was calculated. Experiment was performed in three biological replicates and each replicate was observed under microscope in two technical replicates as described above.

**DNA quantification**. DNA concentration was determined in individual *B. subtilis* swarms and at swarm meeting points of kin or non-kin strains. Pairs of strains were inoculated parallel to each other on semisolid B medium and incubated overnight as described above. Twenty samples were taken at the swarm meeting points and within each swarm with cut pipette tips (Supplementary Fig. 4). Samples were first resuspended in 800 μl Gel Solubilization Buffer and lightly mixed. Samples were heated for 20 min at 50 °C for the agar to melt. Next, they were centrifuged at 10,000 × *g* for 5 min and filtered through a 0.2-μm pore Millipore filter (Merck, KGaA, Germany) previously wetted with 1 ml of 0.9% NaCl solution, to remove the cells. DNA was isolated using PureLink® Quick Gel Extraction Kit (Thermo Scientific, ZDA). Further DNA isolation process was performed as specified in PureLink® Quick Gel Extraction Kit manual. DNA was eluted with 330 μl 1x TE buffer and 100 μl of DNA samples were pipetted into a 96-well microtiter plate where DNA concentration was determined using QuantiFluor® dsDNA System (Promega Corporation, ZDA) according to the manual. The DNA concentration in the agar was calculated using calibration curve obtained from DNA isolation of Lambda DNA standard (QuantiFluor® dsDNA System) added to the agar plate in concentrations: 0 pg/μl, 10 pg/μl, 50 pg/μl and 100 pg/μl. Experiment was performed in three biological replicates and the DNA was measured in three technical replicates.

**DNA quantification and DNA exchange in the presence of exogenous DNA**. To test whether DNA at the meeting point gets digested by *B. subtilis* nucleases exported to the media we measured DNA concentration in plates with/without added DNA at the centre of the plate (Supplementary Fig. 5) where PS-216 (*amy*:p43-*cfp* (Sp) and PS-196 (*sacA*:p43-*yfp* (Cm) swarms met. Approximately 30 μg of DNA (PS-216 Δ*epsA-O* (Tet)) was added to a 1-cm-wide stripe in the middle of the B medium agar plates, where the strains usually meet and 20 samples from the meeting point on B medium plate with/without added DNA were resuspended in 700 μl of 0.9% NaCl instead of Gel Solubilization Buffer from the PureLink® Quick Gel Extraction Kit (Thermo Scientific, ZDA) to prevent damaging viable cells. Sample was centrifuged at 10,000 × *g* for 5 min and filtered through a 0.2-μm pore

Millipore filter (Merck, KGaA, Germany) previously wetted with 1 ml of 0.9% NaCl solution. DNA concentration was determined using QuantiFluor® dsDNA System as described above. Experiment was performed in three biological replicates and the DNA was measured in three technical replicates.

**DNA quantification and DNA exchange between nuclease mutant strains in the presence of exogenous DNA**. To test whether more DNA is found in the agar if nuclease mutant strains are used, and how this affects DNA exchange of the nuclease mutant strains, we simultaneously measured DNA concentration and DNA exchange at the boundary of PS-216 (*amy*:p43-*cfp* (Sp), Δ*nucB* (Kn) Δ*yhcR* (Ery)) and PS-196 (*sacA*:p43-*yfp* (Cm), Δ*yhcR* (Ery)) with/without added DNA at the centre of the plate. DNA quantification and the following procedures are the same as decribed above for wild-type strains. Experiment was performed in three biological replicates and each replicate was sampled in three technical replicates. The DNA concentration was measured in three replicates for each technical replicate.

**Double plate setup for investigating ecological advantage of transformants**. The spread of transformants, cells which acquired the genes from the opposite swarm (*amyE*::p43-*cfp* or *sacA*::p43-*yfp*) on the boundary, that now carry both markers, into area containing two antibiotic markers was tested on B medium agar plate. One half contained semi solid B medium (0.7% agar) and the other half contained semi solid B medium supplemented with two antibiotics: 10 μg/ml of Chloramphenicol (Cm) and 100 μg/ml of Spectinomycin (Sp) (Fig. 5a). First, B media without antibiotics was poured and allowed to solidify, after which one half was carefully removed with a sterile scalpel and B media containing both antibiotics was poured into the plate. Strains (kin and non-kin pairs) were inoculated onto the agar containing no antibiotics and the spread of transformants was observed and sampled after 24 h of incubation at 37 °C at 80% humidity (Fig. 5a). The number of transformants colonising the area containing antibiotics and total CFU was determined by selective plating. In total, 50 colonies carrying both Sp and Cm from three independent experiments were further tested for YFP and CFP fluorescence to confirm the transfer of DNA. Experiment was performed in three biological replicates.

**Relative DNA exchange frequencies in non-kin vs. self**. In order to determine whether DNA exchange frequency of swarming *B. subtilis* PS-216 strain in non-kin combination is significantly higher compared to DNA exchange frequency with an isogenic strain, we used pairs of strains in which one of the strain carried a Δ*comGA* mutation and was not able to accept DNA. This enabled us to determine DNA exchange frequency of only one strain in a pair (kin or non-kin pair) and not total DNA exchange frequency as described above, where transformants of both strains were summed in the equation. Pairs of strains (PS-216 (Sp) was staged with PS-216 Δ*comGA* and PS-196 (Cm)) was staged with PS-196 Δ*comGA*) were inoculated parallel to each other on semisolid B medium, incubated overnight and sampled as described above. DNA exchange frequency was calculated as the number of total CFU/ml of transformants (Sp + Cm) divided by CFU count of the strain not carrying the *comGA* mutation. Experiment was performed in three biological replicates and each replicate was sampled in three technical replicates.

**Scanning electron microscopy**. For observation of the bacterial surface, the selected pieces of colonies were cut from agar plates and fixed in 1% glutaraldehyde and 0.5% formaldehyde in 0.1 M cacodylate buffer, pH 7.3 at 4 °C overnight. After washing of the fixative with 0.1 M cacodylate buffer, the samples were postfixed in 1% aqueous solution of OsO₄ for 1 h. Postfixed samples were dehydrated in an ascending ethanol series (30, 50, 70, 90 and 96%) and transferred into pure acetone that was gradually replaced by hexamethyldisilazane (HMDS) and allowed to air-dry overnight. Dried samples were attached to metal holders with silver paint, coated with platinum and observed with a JEOL JSM-7500F field-emission scanning electron microscope.

**Fitness calculations**. The boundary between non-kin strains PS-216 *amyE*::P43-*cfp* (Sp) and PS-196 *sacA*::P43-*yfp* (Cm) strains was sampled and CFU of each strain was determined in the boundary (CFU_{216non-kin}, CFU_{196non-kin}) and total CFU of both strains (CFU_{non-kin(216+196)}) was determined. Each strain was also staged with a self strain carrying a different Ab marker and CFU of each strain was determined (CFU_{216kin}, CFU_{196kin}), and total CFU was determined (CFU_{kin(216(Sp)+216(Cm))}, CFU_{kin(196(Sp)+196(Cm))}). Fitness was calculated as relative change in strain frequency in non-kin versus kin setting. Relative change in the frequency of strain PS-216 in non-kin setting versus kin setting is shown in the Equation (1) below.

Fitness (relative change in strain frequency in non-kin vs kin setting) =

$$(\mathrm{CFU}_{(216non-kin)}/\mathrm{CFU}_{(non-kin(216+196))})/(\mathrm{CFU}_{(216kin)}/\mathrm{CFU}_{kin(216(Sp)+216(Cm))}) \quad (1)$$

Calculations were performed for data of three biological replicates each in three technical replicates.

***sigW*-YFP expression at the meeting area**. Envelope stress response was observed at the swarm meeting points using kin, non-kin and isogenic strain combinations using a *yfp* (yellow fluorescent protein) gene linked to the promotor of *sigW*, thus cells under envelope stress produced YFP and glowed yellow. Strains were inoculated in parallel on semisolid B media as described above. Fluorescence was observed under stereomicroscope (Leica Microsystems, Germany). Pictures were taken under stereomicroscope of each plate and experiment was performed in at least three biological replicates.

**Statistics and reproducibility**. Data was presented as a mean of three biological replicates ±standard deviation (error bars) or as individual data points (9 technical replicates–3 individual replicates of three biologically independent experiments). Biological replicates were used to calculate mean and sample standard deviation ($n = 3$). For statistical comparison, we performed two tailed Student's *t*-test for unpaired data assuming equal variances. Exact $p$ values are stated in Results. Statistically significant ($p < 0.05$) difference between strain combinations is designated by "*". Mean, standard deviation and t-test calculations were performed using Microsoft Excel 12.00 (2016, Microsoft, Redmond, WA, USA).

**Bioinformatics**. Raw reads for each isolate were obtained after sequencing of dual-indexed Nextera XT libraries (Illumina, USA) prepared from single isolates DNA extracts. Sequencing of the pooled libraries was performed using an Illumina MiSeq benchtop sequencer and $2 \times 250$ bp paired-end reads MiSeq v2 kit-500 cycles reagents (Illumina, USA). Raw reads were trimmed of remnant sequencing adaptors sequences and low-quality regions (Phred score < Q20) using CLC Genomics Workbench 9 (https://www.qiagenbioinformatics.com/). Assembly was performed on adaptors and quality trimmed reads with CLC Genomics Workbench 9 assembler using default parameters.

**Genome analysis**. The degree of relatedness between the 6 *B. subtilis* isolates was evaluated using Average Nucleotide Identity[48], using BLAST similarity scores of 1020 bp long genome segments between all genome pairs (ANIb) as described by Goris et al.[49]. The ANI computation was run using FASTani[17]. Genome sequences are available in the NCBI database under genome accession numbers VBRL00000000, VBRM00000000, VBRN00000000, VBRO00000000, VBRQ00000000 and VBRR00000000.

**Reporting summary**. Further information on research design is available in the Nature Research Reporting Summary linked to this article.

## Data availability
The data that support the findings of this study are available from the corresponding author upon reasonable request and in Source Data file. Genome sequences are available in the NCBI database under genome accession numbers VBRL00000000, VBRM00000000, VBRN00000000, VBRO00000000, VBRQ00000000 and VBRR00000000. Source data are provided with this paper.

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

## Acknowledgements

This work was supported by grants from Slovenian Research Agency (ARRS): the Programme Grant P4-0116, the Slovenia-USA collaboration grant bilateral ARRS project US/18-19-091 and ARRS projects: J4-9302, J4-8228, J4-7637 and the University infrastructural centre "Microscopy of biological samples" located in Biotechnical faculty, University of Ljubljana.

## Author contributions

P.S. and I.M.M conceived the project, P.S., K.B. and R.K. performed the experiments. P.S., J.K., J.N., S.S. and J.M. performed the genome sequencing and assembly analysis. P.S., B.K., R.K., J.N. and J.M. contributed with the methods and P.S. produced the figures. P.S. and I.M.M. supervised the experiments. P.S., M.V. and I.M.M. wrote the manuscript, with all authors contributing to the final version.

## Competing interests

The authors declare no competing interests.
