## [Peer Review File · Nature Communications]

Reviewers' comments:

Reviewer #1 (Remarks to the Author):

What was already known:

These and other authors have recently discovered that many strains of *Bacillus subtilis* can kill other strains when they encounter them on an agar plate or in broth. A number of genes have been identified that contribute to this 'kin discrimination'. *B. subtilis* strains can also develop competence for natural transformation, controlled by a quorum-sensing autoinducer whose specificity can differ between strains.

What the authors did:

They examined transformation frequencies and competence induction at boundaries between strains carrying different genetic markers. The strain that was killing the other also induced its competence genes and recombined DNA from the other strain into its genome. In a selective situation where this recombination increased fitness, the recombinants increased in frequency.

What they concluded:

That the induction of competence at kin-killing boundaries is an important evolutionary mechanism of "promiscuous but safe sex".

Quality of the experimental work:

Very good.

Quality of the presentation (writing, figures):

Very good.

Quality of the interpretation:

Not good.

Throughout, the authors try to achieve generality and significance in two ways, by overgeneralizing and by using anthropomorphic terms. The problems are evident in the provocative title ("Intra-species DNA exchange: *Bacillus subtilis* prefers sex with less related strains"), but they recur many times throughout the manuscript.

First, the process being studied isn't DNA exchange but one-way transformation. In the rest of the manuscript the term 'exchange' is used more than 50 times in reference to transformation, almost always inappropriately. Individual transformation events are not reciprocal exchanges but unidirectional transfers. In the context of this work, all the transfers are unidirectional, since DNA is only taken up by the dominant strain.

'prefers' would only be true if induction of competence during killing of less-related strains provides complete evidence of all factors affecting recombination substrates.

The term 'sex' assumes that the function of DNA uptake is sexual (generating recombinant genotypes). The authors pay lip service to alternative explanations in the first paragraphs of the Introduction and Discussion, but otherwise use 'sex' and 'sexual' as interchangeable with 'competence', 'DNA uptake' and 'transformation'.

More generally, 'social' and other anthropomorphic terms are also frequently misused to increase the apparent significance of the findings.

As is all too common when microbiologists write about natural selection, the manuscript focuses entirely on possible benefits and ignores the costs. For example, one obvious problem with taking up DNA from cells you have just killed, is that you risk getting the target cell genes/alleles that made them sensitive to your attack. In the lab it's easy to counter this by putting the cells on plates that select for acquired antibiotic-resistance genes, but in the real world the counterbalancing advantages are purely speculative.

Overall significance of the findings:

Moderate.

Line-specific points:

36: No. This first sentence is entirely contradicted by the rest of the paragraph.

36: The term 'sexuality' has so many different meanings that it needs to be very specifically

defined, especially when applied to bacteria.

45: But look at Julia Graham and Conrad Istock 1978, and citing work.

47: The quorum-sensing system referred to here is not specifically a competence regulator – it controls many post-exponential processes. And it does not produce sexual isolation, since *B. subtilis* cells take up DNA from any source and grow in environments containing DNAs from many kinds of bacteria.

53: Has *B. subtilis* ever been shown to swarm in a natural environment, rather than on the very unnatural environment of an agar plate?

56: The term 'spitefully' is not used in the cited paper. It's trendy to apply human-behavior terms to interactions between bacteria. But this has led to a lot of over-interpretation...

60: How does 'less than 99.4% nucleotide identity of four housekeeping genes' compare with other measures of identity? I wouldn't call this 'non-kin'!

64: The reference to 'the *Bacillus subtilis* KD system' is misleading. Many genes can contribute to the death of cells at the boundary where non-identical *B. subtilis* strains meet, with different combinations identified in different strain pairs, but this doesn't mean there is a specific 'Kin Discrimination system'. In particular, the use without explanation of the abbreviation 'KD' gives the reader the erroneous impression that genes causing harmful effects to related strains constitute an integrated cellular system.

71: I think the hypothesis is 'higher recombination between less-closely related strains than between more-closely related strains'. Perhaps it could be rephrased.

71: This is an odd hypothesis. Was it developed before the research was done, or after seeing the results?

75-78: This is a very sweeping generalization, based on a simple antibiotic-resistance selection experiment.

80-84: The authors are quite coy here, and don't tell the reader anything about specific kin-killing differences between the strains. This information is highly relevant to understanding the specific killing and transformation events described in the rest of the Results.

81: Are these 'wild type' strains or wild strains?

82: I see that 'average nucleotide identity' doesn't actually mean average nucleotide identity over the genome. The meaning of the measured ANIs should be explained here for the non-expert reader, and some other measure of strain similarity provided for comparison. For example, how different are the gene contents of these strains?

93: Competence is known to differ dramatically in different strains of *B. subtilis* as in other naturally competent bacteria. How well does each strain transform when treated by the standard induction protocol and given purified DNA?

167: Figure 3 contains only 4 data points (2 in each of its panels). There could easily be provided in the text.

188: The term 'stress' is very vague and should be avoided except where it is specifically qualified, e.g. 'cell envelope stress'.

194: The logic underlying this hypothesis is not clear to me.

194: Fig. 4b: This is not DNA exchange frequency, it's transformation frequency. And the strain being transformed isn't specified – is it PS-216?

205: The ~10-fold upregulation is significant, but still only about 1% of cells are turning this gene on.

224: 'aids adaptation' is a gross over-generalization. It's extremely easy to contrive lab conditions that select for transformants. This should never be taken as general evidence that a process exists because it promotes adaptation in the natural environment.

224: I don't see any justification for the word 'ecological'.

254 on: I'm not going to bother making specific comments about the Discussion. It's rife with overgeneralizations and spin.

Reviewer: Rosie Redfield, University of British Columbia

Reviewer #2 (Remarks to the Author):

This manuscript presents the discovery that two *B. subtilis* strains whose relatedness is distant enough such that they can kill/inhibit each other in swarm assays also induce competence in each other to enable horizontal gene transfer. Interestingly, the authors find that the rates of horizontal gene transfer between non-kin strains are higher than between closely related strains. These findings should be broadly interesting to microbiologists and evolutionary biologists.

The authors then use mutant strains to provide evidence that in their swarm assays, competence between non-kin strains is induced by the envelope stress response mediated by sigW, likely as a consequence of contact with antimicrobial compounds at the swarm boundary.

Despite these interesting observations and conclusions, there are substantial problems with this manuscript, which I am listing as major issues below. The authors should address these issues to make the work convincing.

Major issues:

1. Results lines 109-134 (and Figure 2): How is it possible that an increased concentration of extracellular DNA did not affect the transfer frequency? It seems like there is still a major unknown process in the system under investigation. Perhaps it is not the DNA of lysed cells that is used for transfer? Or do the authors have alternative hypotheses? I think the observation that increased concentration of extracellular DNA did not affect the transfer frequency needs to be explained in order to make one of the main points of this paper convincing (i.e. that cell lysis releases DNA that is taken up by non-kin). How can the authors be sure that it is really the DNA of lysed cells that is taken up?
2. Results lines 159-177 (and Figure 3): The authors perform experiments on only one particular non-kin strain combination (PS-216 & PS-196) and then apparently generalize their conclusions for this particular pair to other strain combinations without experimental basis. The authors need to show that transformation primarily occurs in one of the two strains for other non-kin strain combinations.
3. Figure 4: This is only shown for 1 of the 3 combinations of non-kin interactions. If the authors want to really demonstrate that this is an important phenomenon, this observation should be confirmed for all 3 non-kin combinations.
4. Figure 4a: the images are poorly quantified. There needs to be a color bar with numbers. Where is the contact zone for the PS-196 self and the PS-216 self images? Why are there long lines of high YFP signal even in the self-interacting images? Overall, it seems to me that there is a higher density (signal per area) of the YFP reporter at the contact zone of the PS-196 self image, compared to areas that are not in the contact zone. How do the authors explain this?
5. I suggest to completely re-write the abstract, because of the following series of problems:
 - 5a. Title/Abstract/throughout: I understand that "sex" sells, but why not use the term that is most commonly used in the microbiology literature ("horizontal gene transfer"), which is more appropriate for most uses of the word "sex" in the manuscript, except for those uses where the authors appeal to concepts that were developed in evolutionary biology for sex. There are also significant conceptual differences between eukaryotic sexual reproduction and bacterial horizontal gene transfer, of which the authors are of course aware. Rebranding bacterial DNA exchange as "sex" altogether seems like spin/salesmanship and sensationalist language to this reviewer, which should be avoided. In line 226 of the manuscript, the authors described "sex-like benefits" which is a much more down-to-earth description of the process and benefits, and I suggest the authors adhere to such language.
 - 5b. After the first two sentences of the abstract, the sentence structure of the abstract appears to be a collection of statements which only follow a slightly coherent logical order, unfortunately. (eg: how do the authors arrive at "sexual isolation" in the 3rd sentence? Which dogma do the authors refer to?).
 - 5c. Please also clearly state what the hypothesis or goal or motivation of the study is – the fact that "DNA exchange between two interacting *B. subtilis* strains has never been addressed previously" is only a motivation for descriptive work and does not reflect the interesting contents of the paper very well.
 - 5d. "DNA exchange between two interacting *B. subtilis* strains has never been addressed previously". Careful with such claims of priority.

5e. "... which is in contrast to the current dogma". This claim seems sensationalist. Also: which current dogma?

5f. Why is the horizontal gene transfer process described here as "safe sex"? Such anthropomorphic statements seem like inaccurate spin/salesmanship to me. This should be avoided.

5g. Last sentence: "it is possible that this (or similar) mechanisms..." This statement is purely speculative without any evidence, and therefore I think it should not be part of the abstract as a concluding statement.

Issues that are also important:

6. Introduction: There is a contradiction in the way the previous literature is introduced, yet this contradiction is not discussed: How is it possible that two non-kin *B. subtilis* strains induce each other's competence to take up foreign DNA, even though previous literature shows that only members of the same mating group/pherotype can induce each other's competence (Refs 9-11)? I understand that the observation that non-kin strains can induce each other's competence through sigW is a main discovery of the manuscript, but somehow the introduction does not mention this contradiction.

7. What is the relative importance of sigW/stress-related competence induction and quorum sensing mediated competence induction – please comment in the discussion at least.

8. I think that adding supplementary Table 1 into the main manuscript would significantly help the readership of this manuscript, as it is otherwise difficult to keep track of strain names and their relatedness/kin-groups.

9. Discussion: Line 253: Why is the role of competence and DNA update controversial? It does not seem controversial to me.

10. Two paragraphs in the discussion (lines 294-309 and lines 310-321) contain purely speculative ideas. These are interesting, but have no experimental basis. The authors use the language "it is possible that...". Many things are possible, of course. I suggest the authors write "we speculate that...".

11. Last sentence in the discussion: "promiscuous but safe sex" This is an oversimplification of the authors' results that is more like a newspaper article jargon. What is the definition of "safe" in this context and "promiscuous"? Statements like this should be avoided.

Minor issues:

12. Discussion, line 275-276: change "bacteria"  "*B. subtilis*". The authors have not shown this for all bacteria, only for *B. subtilis*.

13. Generally, I don't think it is necessary to keep carrying the complete genotype description (amyE::etc) for each strain at each location in the text. This makes it difficult to read. I suggest: define your strain names somewhere and then stick to those names.

14. Results, first paragraph: "genome sequences of the six wild type *B. subtilis* strains...". Why is it "the" six strains? Why these strains? Why six? I understand that some of this information is in the methods section, but some explanatory note needs to go into the results section as well to motivate the strain usage.

15. Figure 1a: I presume these are representative images of a particular strain combination. This should be stated in the caption. The caption for Figure 1a is too short. Why are there hand-written notes on the petri dishes? Are readers supposed to read them (I can't clearly decipher them)?

16. Figure 1b: please show all individual data points from which the error bars were calculated. Knowing CFU counts, and especially ratios of CFU counts, I am very surprised by the very small error bars. What do the error bars represent?

17. Figure 1b: caption. Please check brackets.

18. Figure 2b: please do not abbreviate strain names, to stay consistent with strain names (which are already a bit confusing).

19. Figure 2, 3, caption: the abbreviation KD is never defined. I presume it means kin discrimination.

20. Figure 3: please show individual data points. For the PS-196 strain there are no error bars.

21. Figure 3: Why is PS-13 in the caption but not in the figure itself?

22. Please define ANI on first use.

23. Title: Why the review-like title before the ":"? The content of this review-like statement is

implicit in the second part of the title.

24. Title: Something went wrong with the italics in the title.

Reviewer #3 (Remarks to the Author):

The authors report that the encounter of non-kin cells stimulates DNA exchanges by transformation in *Bacillus subtilis*. They provide evidence that this occurs through killing of non-kin cells and incorporation of the released DNA. As stated by the authors, killing by non-kin and subsequent import of their DNA has been documented in other species. So, this is not conceptually novel. Yet, it is the first that this is described in *B. subtilis*, suggesting that non-kin predation for the purpose of getting DNA is conserved in naturally transformable bacteria. The authors have extensively documented the emergence of recombinants from two non-kin strains at the boundary between the strains swarms. It is interesting that the mechanism of kin-recognition and killing is likely distinct from other documented situations. However, this is mostly left unresolved and the authors are more focused on the evolutionary implications. To support their hypothesis (transformation as a means of genetic mixing), they provide a nice illustration that the recombinants produced at the boundary between non-kin cells could colonize a section of a plate on which the parents strains could not grow. The nature of the experimental system (swarm plates) makes it difficult to address some key points and I am not fully convinced of what is actually going on. Below are some of my concerns:

- The quantification of DNA released at the boundary is technically challenging and I found these experiments inconclusive. The assay in fig 2b technically did not differentiate free DNA released by the dead cells from total DNA contained in the intact cells within the swarm. So, there is no point in comparing the amount of DNA at the boundary and within the swarm. Indeed, the DNA at the boundary may be short-lived because of cytoplasmic DNAses released by the dead cells. In this respect, the role of "secreted" DNase is also not clear. The amount of DNA at the boundary is compared between two WT and between two mutant strains which both carry mutations for two nucleases (*yhcR* and *nucT*, and which are not described in the text). It then cannot be known if the decrease in DNA at the boundary originates from DNase secretion by PS-216 or from the release of DNase by the dead PS-196.

But most of all, what I found puzzling is that addition of DNA did not affect DNA exchange. This is not consistent with the involvement of natural transformation.

- *comGA* mutants were used to determine the directionality of the DNA exchange. *ComGA* has been found to be required for natural transformation but plays additional roles (pilus assembly, persistence, inhibition of cell elongation). The use of a *comEC* mutant would have been more convincing.

- The authors claim that import of DNA from non-kin cells is a consequence of competence induction caused by the non-kin encounter (line 25, 74, 170, 208). I find the evidence supporting this claim rather weak. *ComGA* is indeed required, supporting a role for natural transformation. However, I do not see evidence for up-regulation of competence genes, or the induction of the competence state. Technically, the only data available do not show up-regulation of *comGA*, but rather a ~10-fold increase in the number of *ComGA-yfp* positive cells. There is clearly a basal fraction of cells expressing *ComGA-yfp* in the swarm, probably in the competence state. Competent cells of *B. subtilis* have been shown to be tolerant to antimicrobials (PMID: 25899641), the mechanism proposed to be involved in non-kin killing. So an increased number of *ComGA-yfp* may be due to the tolerance of the competent cells to the antimicrobial produced at the boundary, not because *comGA* is induced in a higher number of cells. The regulation of competence in *B. subtilis* is well established and I guess there might be some mutants that could be used to validate/invalidate the induction of competence genes at the boundary.

- The link between sensing of membrane damage and induction of competence is not clear. From figure 4a, *sigW* is mostly induced at the boundary by the "prey" strain (PS-196) but

transformation occurs mainly in the strain in which sigW is not visibly induced, PS-216. How is this explained? Does secretion of competence pheromones by PS-196 trigger competence in PS-216? If not, please provide direct evidence that sigW induction triggers competence in PS-216.

Other comments:

- Student's T test are extensively used all along the manuscript. Unless the authors can show that the data follow a normal distribution, student's T test are inappropriate. Please seek advice from a statistician.
- Line 64. Please define KD (kin-discrimination, I suppose)
- Line 82-84. It would just have been easier for the reader to refer to the 3 recognition types previously described (3, 7, 9)
- Fig 2b. Please add a legend to the y-axis
- Line 163. The original reference showing that ComGA is required for transformation is Chung and Dubnau, 1998. PMID: 9422590.
- Line 172. "less-competent strain". There is no data showing that the strain that generates the fewer recombinant is actually less competent. Have the authors determined the ability of each the two strains to transform using added DNA?
- Line 270-275. Please consider breaking this long sentence into 2-3 sentences.
- Line 288/290. There is no data in this manuscript showing that the tested strains have different phenotypes.
- Line 328. The finding could also be consistent with the "genome-curing hypothesis" (Croucher, Plos Biol, 2016, PMID: 26934590). This hypothesis is never mentioned and should be discussed.

Dear reviewers,

We hereby provide a rebuttal for our (re-titled) paper invited for resubmission “Kin discrimination promotes horizontal gene transfer between unrelated strains in *Bacillus subtilis*”. We thank the reviewers for their exhaustive comments which have greatly improved the manuscript (point-by-point responses to all comments below). Changes to this new version are extensive and can be summarised by the following three points:

First, in response to reviewers’ comments we have added new experimental **data in the manuscript**:

- 1) *Experiment sigW activation (Fig. 5a and supplementary figure 5) – all three non-kin combinations, all kin and all self combinations.*
- 2) *ΔsigW DNA transfer (Fig 5b) – all three non-kin combination*
- 3) *Activation of comGA (Fig 5d) – two non-kin and one more kin combination. The reason the last combination PS-13:PS-196 was not tested is that the reporter strain was PS-216-comGA-YFP. For comparison we added an additional kin strain combination.*
- 4) *ΔsigW DNA transfer between wt PS-216 and ΔsigW PS-196 mutant (Fig 5c)*
- 5) *Activation of comGA in PS-216 when paired with ΔsigW PS-196 mutant (Fig 5d)*
- 6) *Adaptation experiment (Supplementary figure 6 and 7) for all three non-kin combinations, all kin and all self combinations*

We have also added **data in the rebuttal** (Rebuttal Figures R1-3). Specifically, we have performed an additional experiment to test the validity of using *comGA* versus a *comEC* mutant strain, as requested by the reviewer 3 (Figure R2). We furthermore show unpublished data in the rebuttal in response to reviewer 1: a dendrogram showing the relationship between KD genes and phylogeny (Figure R1) and a table showing the distribution of KD genes across the 6 strains (Table R1). We also include transformation frequencies of strains in liquid monocultures (Table R2) and CFUs of transformants at the meeting areas and cell counts of reporter strain PS-216 comGA-YFP at kin and non-kin encounters (Figure R3).

Second, we have changed the structure of the paper, including deemphasising the results on free DNA concentration, shortening the Discussion section and revising all figures.

Third, we have completely revised the text, and changed terminology throughout as requested by Reviewers 1 and 2. Please find our explanations to the reviewers’ comments in *blue text* below.

Reviewer #1 (Remarks to the Author):

What was already known:

These and other authors have recently discovered that many strains of *Bacillus subtilis* can kill other strains when they encounter them on an agar plate or in broth. A number of genes have been identified that contribute to this ‘kin discrimination’. *B. subtilis* strains can also develop competence for natural transformation, controlled by a quorum-sensing autoinducer whose specificity can differ between strains.

What the authors did:

They examined transformation frequencies and competence induction at boundaries between strains carrying different genetic markers. The strain that was killing the other also induced its competence genes and recombined DNA from the other strain into its genome. In a selective situation where this recombination increased fitness, the recombinants increased in frequency.

This is an accurate breakdown of our results and we would just like to add that we also could show that transformation rate was higher between genomically less related strains, which seemed counterintuitive at first, as the efficiency of transformation-mediated recombination of more divergent DNA fragments has been shown to decrease log-linearly with increasing sequence dissimilarity in Bacillus (Roberts & Cohan, 1993, Zawadzki et al., 1995, Majewski, 2001). This demonstrates that social interactions leading to competence development can override constraints imposed by sequence similarity in taking up foreign DNA.

What they concluded:

That the induction of competence at kin-killing boundaries is an important evolutionary mechanism of “promiscuous but safe sex”.

Quality of the experimental work:

Very good.

Quality of the presentation (writing, figures):

Very good.

Quality of the interpretation:

Not good.

Throughout, the authors try to achieve generality and significance in two ways, by overgeneralizing and by using anthropomorphic terms. The problems are evident in the provocative title (“Intra-species DNA exchange: *Bacillus subtilis* prefers sex with less related strains”), but they recur many times throughout the manuscript.

We agree that there were problems with some of the terminology we used and have made major changes throughout the manuscript. We now do not use the term ‘sex’ for instance. We will detail these changes in individual points below.

First, the process being studied isn’t DNA exchange but one-way transformation. In the rest of the manuscript the term ‘exchange’ is used more than 50 times in reference to transformation, almost always inappropriately. Individual transformation events are not reciprocal exchanges but unidirectional transfers. In the context of this work, all the transfers are unidirectional, since DNA is only taken up by the dominant strain. ‘prefers’ would only be true if induction of competence during killing of less-related strains provides complete evidence of all factors affecting recombination substrates.

We appreciate this comment and agree that on the level of individuals, cells are either donors or recipients and the process is unidirectional and not bidirectional. We note that on the level of staged swarms, we observe both DNA donation and DNA uptake and recombination within a single genotype (see Fig 1) and so it could be classified as exchange on the population level. To avoid confusion, we have removed the use of exchange from the manuscript as recommended by the reviewer. We have also rephrased sentences containing the verb ‘prefers’.

The term 'sex' assumes that the function of DNA uptake is sexual (generating recombinant genotypes). The authors pay lip service to alternative explanations in the first paragraphs of the Introduction and Discussion, but otherwise use 'sex' and 'sexual' as interchangeable with 'competence', 'DNA uptake' and 'transformation'. More generally, 'social' and other anthropomorphic terms are also frequently misused to increase the apparent significance of the findings. As is all too common when microbiologists write about natural selection, the manuscript focuses entirely on possible benefits and ignores the costs. For example, one obvious problem with taking up DNA from cells you have just killed, is that you risk getting the target cell genes/alleles that made them sensitive to your attack. In the lab it's easy to counter this by putting the cells on plates that select for acquired antibiotic-resistance genes, but in the real world the counterbalancing advantages are purely speculative.

As noted above, we have reworded many phrases throughout the manuscript. We have greatly shortened the Discussion section, including passages elaborating on 'sex-like' benefits. We agree that the adaptive benefits of recombination are not straightforward to demonstrate and we appreciate that all explanations of bacterial recombination (adaptation, repair and food) are still on the table (and are non-mutually exclusive). We do note (in the new Discussion) that (line: 273-276): "The observed increase in recombination between genomically more distant strains is not predicted by the DNA for food hypothesis and poses a problem for the DNA repair hypothesis, but would be consistent with the sex hypothesis."

We now note in the Results that the experiment in Fig. 6 is 'proof of concept' only (line 221-222) and we end the manuscript with the sentence (lines 283-285) "Evolution experiments incorporating ecological realism are needed to shed light on the adaptive benefits of KD-mediated antagonism and transformation."

Overall significance of the findings:
Moderate.

Line-specific points:

36: No. This first sentence is entirely contradicted by the rest of the paragraph.

This sentence has been removed from the manuscript.

36: The term 'sexuality' has so many different meanings that it needs to be very specifically defined, especially when applied to bacteria.

This term has been removed from the manuscript.

45: But look at Julia Graham and Conrad Istock 1978, and citing work.

This is correct and we now state:

Line no. 52-54:

"B. subtilis is naturally competent^{10,11} but transformation has mostly been studied in the context of single clones growing in liquid medium in this species (but see¹²⁻¹⁴), precluding the action of social interactions such as swarming found in structured environments."

47: The quorum-sensing system referred to here is not specifically a competence regulator – it controls many post-exponential processes. And it does not produce sexual isolation, since *B. subtilis* cells take up DNA from any source and grow in environments containing DNAs from many kinds of bacteria.

Although it has been reported that QS serves as a behavioural mechanism of sexual isolation in B. subtilis and other species (Cohan, 2002, Tortosa & Dubnau, 1999, Solomon & Grossman, 1996, Ansaldi & Dubnau, 2004), we have modified this passage and no longer use the term ‘sexual isolation’ for QS.

53: Has *B. subtilis* ever been shown to swarm in a natural environment, rather than on the very unnatural environment of an agar plate?

Swarming has not been filmed/observed in a natural setting to our knowledge, but evidence exists that genes required for swarming are important for colonization of roots in Bacillus subtilis (Bais et al., 2004, Gao et al., 2016).

56: The term ‘spitefully’ is not used in the cited paper. It’s trendy to apply human-behavior terms to interactions between bacteria. But this has led to a lot of over-interpretation...

This sentence has been removed from the manuscript.

60: How does ‘less than 99.4% nucleotide identity of four housekeeping genes’ compare with other measures of identity? I wouldn’t call this ‘non-kin’!

‘Kin’ and ‘non-kin’ are defined here as recognising self and non-self respectively during swarming on agar, as in previous studies (e.g. (Stefanic et al., 2015)). Instead of the four housekeeping genes used in our previous study based on which we determined the 99.4% nucleotide identity threshold for non-kin, we here used entire core genomes to calculate average genomic nucleotide identity in this study, which is the highest resolution measure available. We have now added nucleotide identity (in addition to ‘self’, ‘kin’ and ‘non-kin’ designations) to Figure 1.

64: The reference to ‘the *Bacillus subtilis* KD system’ is misleading. Many genes can contribute to the death of cells at the boundary where non-identical *B. subtilis* strains meet, with different combinations identified in different strain pairs, but this doesn’t mean there is a specific ‘Kin Discrimination system’. In particular, the use without explanation of the abbreviation ‘KD’ gives the reader the erroneous impression that genes causing harmful effects to related strains constitute an integrated cellular system.

For our answer we refer to a study in Current Biology (Lyons et al., 2016) by authors PS and IMM with collaborators at Harvard that details the molecular genetic basis of KD associated genes. We also introduce the abbreviation “KD” at first use (line 33).

71: I think the hypothesis is ‘higher recombination between less-closely related strains than between more-closely related strains’. Perhaps it could be rephrased.

This statement is no longer present in the current manuscript.

71: This is an odd hypothesis. Was it developed before the research was done, or after seeing the results?

This relates to the now deleted sentence “We here test the hypothesis that antagonistic cell-interactions between two non-kin B. subtilis strains during swarm boundary formation result in increased recombination between less related strains.” We do not think this hypothesis is odd, as a relationship between strain-specific killing coupled to transformation has been demonstrated in other species before. We now state in the introduction:

Lines 46-51:

Recognition and lysis of neighbouring genotypes via the release of effectors by the T6SS secretion system coupled to natural transformation has been demonstrated in the gram-negative species Vibrio cholerae⁵ and Acinetobacter baylyi^{6,7}. In the gram-positive species Streptococcus pneumoniae, bacteriocin release can result in lysis of neighbouring susceptible genotypes and likewise increase transformation-mediated horizontal gene transfer^{8,9}.

75-78: This is a very sweeping generalization, based on a simple antibiotic-resistance selection experiment.

This refers to the now deleted statement “Moreover, we show that gene exchange between non-kin can, through increased genetic variation, increase the rate of adaptation and that horizontal gene transfer between non-kin swarms results in the form of successful exploitation of novel adaptive zones.” Apart from deleting this passage, we now describe this simple antibiotic-resistance selection experiment as a “proof-of-principle experiment” (Line 221). Furthermore we now state in the Results:

Lines 217-221:

“The evolutionary benefits of competence and transformation are still the subject of active debate and several, non-mutually exclusive benefits have been proposed including nutrient acquisition, repair and sex-like benefits^{25,26}. As the uptake and recombination of extracellular DNA is an integral part of transformation, it is often assumed that recombination-based benefits are at least partly responsible for its evolutionary maintenance²⁵.” which we believe is not controversial.

80-84: The authors are quite coy here, and don't tell the reader anything about specific kin-killing differences between the strains. This information is highly relevant to understanding the specific killing and transformation events described in the rest of the Results.

In a previous study (Lyons et al., 2016) we uncovered genetic mechanism underlying KD behaviours. Genes involved in KD phenotype belonged to cell-surface modification and antimicrobial production and response. The KD genes varied significantly in expression level and mutation phenotype among B. subtilis strains, suggesting interstrain variation in the exact kin discrimination mechanism used. Genome analyses on B. subtilis strains deposited in the databases (and not the ones used in the study) revealed a substantial diversity of antimicrobial genes present in unique combinations in different strains. The current study employs different B. subtilis strains using the same KD mechanisms, although the precise makeup of effectors in these strains will be different. A detailed characterisation of the specific KD gene combinations in these strains is beyond the scope of this study but below we present preliminary evidence that the core genome phylogeny is congruent with a phylogeny based on putative KD genes found in the same 6 strains used in this study (Figure R1, unpublished results) and how presence/absence of KD genes corresponds to KD groups (Table R1).

Figure R1: Clustering of KD genes and phylogeny of the 6 *B. subtilis* strains used in this study. Left: Clustering of putative KD genes that were present in all strains (genes included: xh1A, xh1B, ytnP, bacC, bacE, ppsE, srfAD, srfP_dhbC, ntdB, pksF, bpsA, ytpA, ytpB, ndoA, ndoAI_spolISA, spollSB, yitM, yitQ; 20403 bp). Right: phylogenetic tree based on housekeeping genes (fabD, fusA, gyrA, gyrB, prfA, rpoB, rpoD, yycF; 13893 bp)

Table R1: Preliminary comparison of presence/absence of KD genes in kin groups.

	PS-13	PS-18	PS-68	PS-216	PS-218	PS-196
yqcG/yqcF	+	+	+	+	+	-
bsrH	+	+	+	+	+	-
bsrE	+	+	+	+	+	-
YobL/YobK	+	+	+	+	+	-
WapA/I	+	+	+	+	-	-
yxiD/yxxD	+	+	+	+	-	-
skfA-H	+	+	+	+	-	-
sdpA-R	+	+	+	+	-	-
txpA/ratA	+	+	+	+	-	-

Differently coloured columns represent distinct kin groups.

81: Are these 'wild type' strains or wild strains?

We use 'wild type' to distinguish unengineered strains from engineered ones (we have added a new Table (2) summarising strains used in this study). All such wild type strains are environmental isolates- we now state "we first tested for swarm boundary formation between all pairwise combinations of six B. subtilis strains isolated from a micro-scale soil population (Stefanic & Mandic-Mulec, 2009) and calculated their degree of relatedness".

82: I see that 'average nucleotide identity' doesn't actually mean average nucleotide identity over the genome. The meaning of the measured ANIs should be explained here for the non-expert reader, and some other measure of strain similarity provided for comparison. For example, how different are the gene contents of these strains?

This is a good point. ANI has become a standard measure of relatedness, and even though it does not strictly represent core genome evolutionary relatedness, it closely reflects the traditional microbiological concept of DNA-DNA hybridization relatedness for defining species (Jain et al., 2018, Goris et al., 2007). Apart from that, it offers several important advantages such as higher resolution among closely related genomes and the fact that relatedness can be estimated among draft (incomplete) genome sequences (encoding at least a few hundred shared genes), greatly expanding the number of sequences that can be

studied and classified compared to a universal gene-based approach (Jain et al., 2018, Konstantinidis & Tiedje, 2005). We added this sentence to the manuscript:

Lines 65-68:

“... we first tested for swarm boundary formation between all pairwise combinations of six *B. subtilis* strains isolated from a micro-scale soil population¹⁵ and calculated Average Nucleotide Identity (ANI¹⁶), which is defined as the mean nucleotide identity of orthologous gene pairs shared between two microbial genomes¹⁷ (Table 1). “

93: Competence is known to differ dramatically in different strains of *B. subtilis* as in other naturally competent bacteria. How well does each strain transform when treated by the standard induction protocol and given purified DNA?

Transformation frequency by standard protocol in monocultures with added DNA is shown in table R2 (unpublished results). We do have data on DNA transfer in liquid co-cultures in the supplemental figures (Supplemental Figure 1c and 1d)

Strain	Average Transformation Frequency	Standard deviation
PS-216	-1,725	0,021
PS-13	-2,695	1,110
PS-18	-4,330	0,552
PS-68	-3,415	0,205
PS-196	-4,090	0,467
PS-218	-3,960	0,170

*Table R2: Average transformation frequencies of *B. subtilis* monocultures with added purified DNA determined by the standard induction protocol.*

167: Figure 3 contains only 4 data points (2 in each of its panels). There could easily be provided in the text.

This Figure is no longer present in the paper.

188: The term ‘stress’ is very vague and should be avoided except where it is specifically qualified, e.g. ‘cell envelope stress’.

We now use ‘cell envelope stress’ or when just ‘stress’ is used this is qualified by mentioning sigW expression or is used in a more general sense.

194: The logic underlying this hypothesis is not clear to me.

This refers to “As expression of σ^W was more prominent at the boundary (Fig. 4a) we hypothesized that an inability to respond to stress (Δ sigW) in non-kin encounters would prevent the induction of competence.” A sentence that is no longer present in the current version of the manuscript.

This is now in the manuscript:

Lines 190-196:

“How can the fact that the dominant strain acts as the recipient of DNA (Fig. 2) be reconciled with the finding that DNA uptake and recombination is associated with a stress response (sigW expression) which can be expected to be most prominent in the lysed (not the lysing) strain (Fig. 2)? We hypothesised that the sigW expression in the lysed strain induces competence in the lysing strain. To test this, we staged the wt PS-216 strain with a PS-196 Δ sigW mutant (the latter strain undergoes lysis in this pairing, (Fig. 3b) and measured transformation rates.”

194: Fig. 4b: This is not DNA exchange frequency, it's transformation frequency. And the strain being transformed isn't specified – is it PS-216?

We now use transformation frequency throughout and do not use “DNA exchange frequency”. The Figure has been reformatted.

205: The ~10-fold upregulation is significant, but still only about 1% of cells are turning this gene on.

*Different wild type strains have different competence levels (see Table R2). We sampled at one time point and 1% is what we detected in that exact time of sampling in strain PS-216. Because competence is transient, it is possible that more than 1% of cells turn the comGA gene on (sooner or later during the interaction). Literature says that 10-20 % of laboratory strain *B. subtilis* 168 turn on competence genes (Mirouze et al., 2012, Dubnau, 1991) and the problem of wild type strains with no transformation ability was addressed previously (Nijland et al., 2010). In the collection of 40 *B. subtilis* strains in our laboratory, only 16 are naturally competent (unpublished) or their competence is below the detection limit.*

224: ‘aids adaptation’ is a gross over-generalization. It's extremely easy to contrive lab conditions that select for transformants. This should never be taken as general evidence that a process exists because it promotes adaptation in the natural environment.

This phrase appears in the revision as (lines 221-224):

“A proof-of-principle experiment was devised to test whether KD-mediated transformation could aid adaptation through the uptake of beneficial alleles originating from the encounter with a non-kin swarm.” We do not claim it proves that transformation is solely selected because of its sex-like benefits but merely that this observation is consistent with it.

224: I don't see any justification for the word ‘ecological’.

This has since been rephrased.

254 on: I'm not going to bother making specific comments about the Discussion. It's rife with overgeneralizations and spin.

We have now completely revised the discussion.

Reviewer: Rosie Redfield, University of British Columbia

Reviewer #2 (Remarks to the Author):

This manuscript presents the discovery that two *B. subtilis* strains whose relatedness is distant enough such that they can kill/inhibit each other in swarm assays also induce competence in each other to enable horizontal gene transfer. Interestingly, the authors find that the rates of horizontal gene transfer between non-kin strains are higher than between closely related strains. These findings should be broadly interesting to microbiologists and evolutionary biologists. The authors then use mutant strains to provide evidence that in their swarm assays, competence between non-kin strains is induced by the envelope stress response mediated by *sigW*, likely as a consequence of contact with antimicrobial compounds at the swarm boundary. Despite these interesting observations and conclusions, there are substantial problems with this manuscript, which I am listing as major issues below. The authors should address these issues to make the work convincing.

Major issues:

1. Results lines 109-134 (and Figure 2): How is it possible that an increased concentration of extracellular DNA did not affect the transfer frequency? It seems like there is still a major unknown process in the system under investigation. Perhaps it is not the DNA of lysed cells that is used for transfer? Or do the authors have alternative hypotheses? I think the observation that increased concentration of extracellular DNA did not affect the transfer frequency needs to be explained in order to make one of the main points of this paper convincing (i.e. that cell lysis releases DNA that is taken up by non-kin). How can the authors be sure that it is really the DNA of lysed cells that is taken up?

The explanation is relatively straightforward and has been better explained in the current version of the manuscript: DNA concentrations are already saturating in the absence of antagonistic interactions and hence lysis in swarm boundaries does not increase transformation rate. Regarding the importance of extracellular DNA, we present multiple findings:

- *Removal by adding DNAses shows extracellular DNA is a prerequisite for transformation (Supplementary Fig. 3).*
- *Extracellular DNA concentrations measured at the boundary between non-kin strains were comparable to those between staged kin- or isogenic strains (Fig. 4a, Supplementary figure 4a).*
- *Extracellular DNA concentrations were significantly higher at the boundary between non-kin nuclease mutant strains compare to wt strains (Fig 4b, Supplementary figure 4b).*
- *Elevated concentrations of extracellular DNA did not significantly affect transformation frequencies: transformation frequency between non-kin nuclease mutants was not significantly different from corresponding wild type strains (Fig. 4c, Supplementary figure 4b).*

2. Results lines 159-177 (and Figure 3): The authors perform experiments on only one particular non-kin strain combination (PS-216 & PS-196) and then apparently generalize their conclusions for this particular pair to other strain combinations without experimental basis. The authors need to show that transformation primarily occurs in one of the two strains for other non-kin strain combinations.

We have added other non-kin combinations to show that transformation occurs primarily in one strain (see figure 2). Apart from PS-216: PS-196, we have now paired: PS-196: PS-13 and PS-218: PS-216 and tested the direction of DNA transfer, which was always in favour of strain PS-216 (paired with PS-196 or PS-218) and PS-196 when paired with PS-13 (Figure 2).

3. Figure 4: This is only shown for 1 of the 3 combinations of non-kin interactions. If the authors want to really demonstrate that this is an important phenomenon, this observation should be confirmed for all 3 non-kin combinations.

Old Figure 4 is now Figure 5. We performed additional experiments for non-kin combinations as requested:

- 1. Experiment sigW activation (Fig. 5a and supplementary figure 5) – all three non-kin combinations, all kin and all self-combinations*
- 2. Δ sigW DNA transfer (Fig 5b) – all three non-kin combination*
- 3. Activation of comGA (Fig 5d) – two non-kin and one more kin combination. The reason the last combination PS-13:PS-196 was not tested is that the reporter strain was PS-216-comGA-YFP. For comparison we added an additional kin strain combination.*

4. Figure 4a: the images are poorly quantified. There needs to be a color bar with numbers. Where is the contact zone for the PS-196 self and the PS-216 self images? Why are there long lines of high YFP signal even in the self-interacting images? Overall, it seems to me that there is a higher density (signal per area) of the YFP reporter at the contact zone of the PS-196 self-image, compared to areas that are not in the contact zone. How do the authors explain this?

We have now added the activation of sigW images of kin and isogenic strains (Supplementary Fig 5), where there is some background of the fluorescence, but it is clear that the activation of sigW occurs in non-kin combinations. We have repeated the experiment in three biological replicates and obtained the same visual results. This was also previously published in our paper Lyons et al (2016) and is just confirmed here visually.

5. I suggest to completely re-write the abstract, because of the following series of problems:

We have completely rewritten the Abstract.

5a. Title/Abstract/throughout: I understand that “sex” sells, but why not use the term that is most commonly used in the microbiology literature (“horizontal gene transfer”), which is more appropriate for most uses of the word “sex” in the manuscript, except for those uses where the authors appeal to concepts that were developed in evolutionary biology for sex. There are also significant conceptual differences between eukaryotic sexual reproduction and bacterial horizontal gene transfer, of which the authors are of course aware. Rebranding bacterial DNA exchange as “sex” altogether seems like spin/salesmanship and sensationalist language to this reviewer, which should be avoided. In line 226 of the manuscript, the authors described “sex-like benefits” which is a much more down-to-earth description of the process and benefits, and I suggest the authors adhere to such language.

We agree and have changed language throughout the paper (and changed the title) (see also comments Reviewer 1).

5b. After the first two sentences of the abstract, the sentence structure of the abstract appears to be a collection of statements which only follow a slightly coherent logical order, unfortunately. (eg: how do the authors arrive at “sexual isolation” in the 3rd sentence? Which dogma do the authors refer to?).

We have completely rephrased the abstract and no longer use the term “sexual isolation” or refer to a “dogma”.

5c. Please also clearly state what the hypothesis or goal or motivation of the study is – the fact that “DNA exchange between two interacting *B. subtilis* strains has never been addressed previously” is only a motivation for descriptive work and does not reflect the interesting contents of the paper very well.

In this revision we write at the end of the Introduction:

Lines 52-56:

*“*B. subtilis* is naturally competent^{10,11} but transformation has mostly been studied in the context of single clones growing in liquid medium in this species (but see¹²⁻¹⁴), precluding the action of social interactions such as swarming found in structured environments. We hypothesized that the antagonisms observed between genetically distinct *B. subtilis* strains could lead to transformation-mediated recombination. We here demonstrate...”*

This provides a link between the examples of competence previously coupled to strain killing in other species and what we find in this study.

5d. “DNA exchange between two interacting *B. subtilis* strains has never been addressed previously”. Careful with such claims of priority.

This is a good point and this sentence has been deleted from this version.

5e. “... which is in contrast to the current dogma”. This claim seems sensationalist. Also: which current dogma?

This phrase has been removed from the current manuscript.

5f. Why is the horizontal gene transfer process described here as “safe sex”? Such anthropomorphic statements seem like inaccurate spin/salesmanship to me. This should be avoided.

This phrase has been removed from the current manuscript.

5g. Last sentence: “it is possible that this (or similar) mechanisms...” This statement is purely speculative without any evidence, and therefore I think it should not be part of the abstract as a concluding statement.

This phrase has been removed from the current manuscript.

Issues that are also important:

6. Introduction: There is a contradiction in the way the previous literature is introduced, yet this contradiction is not discussed: How is it possible that two non-kin *B. subtilis* strains induce each other’s competence to take up foreign DNA, even though previous literature shows that only members of the same mating group/pherotype can induce each other’s competence (Refs 9-11)? I understand that the observation that non-kin strains can induce each other’s competence through sigW is a main discovery of the manuscript, but somehow the introduction does not mention this contradiction.

We have shortened the Introduction and have moved the pherotype issue to the Discussion. We completely agree that the contrast between the pherotype paradigm and our findings are both important and puzzling. We now state:

Lines 248-259:

*“Induction of competence in *B. subtilis* requires activation of a quorum sensing (QS) system, encoded by the comQXPA operon^{30,31}, that ensures activation of the competence genes via the induction of ComS^{32,33}. We and others have shown that *B. subtilis* genotypes evolved extensive polymorphism in the ComQXPA QS system^{15,34-36}. In liquid competence media, competence can only be induced if two strains express the same pheromone (i.e. both strains belong to the same pherotype)^{15,34,37}. Limited cross-talk and sometimes even inhibition of competence has been observed between different pherotypes³⁴, leading to the expectation that transformation between pherotypes is lower than within pherotypes. In contrast, the non-kin strains that engage in transformation in this study all belong to different pherotypes^{2,15}, for example, PS-216 and other kin strains belong to “pherotype 168”, whereas PS-218 and PS-196 belong to “NAF4 pherotype”¹⁵, yet still show higher transformation efficiency when paired than do more closely related kin strain pairs belonging to the same pherotype.”*

This observation is novel and it demonstrates a possibility that different mechanisms operate HGT during surface-bound cell-cell interactions.”

7. What is the relative importance of sigW/stress-related competence induction and quorum sensing mediated competence induction – please comment in the discussion at least.

We have added a section on sigW and quorum sensing mediated competence induction in the discussion Lines: 239-247.

8. I think that adding supplementary Table 1 into the main manuscript would significantly help the readership of this manuscript, as it is otherwise difficult to keep track of strain names and their relatedness/kin-groups.

We have now provided a Table (2) with strain designations/abbreviations.

9. Discussion: Line 253: Why is the role of competence and DNA uptake controversial? It does not seem controversial to me.

We have removed this phrase in the revision (see comments by Reviewer 1 who is critical of the ‘sex hypothesis’).

10. Two paragraphs in the discussion (lines 294-309 and lines 310-321) contain purely speculative ideas. These are interesting, but have no experimental basis. The authors use the language “it is possible that...”. Many things are possible, of course. I suggest the authors write “we speculate that...”.

We agree and these passages are no longer present in the current draft.

11. Last sentence in the discussion: “promiscuous but safe sex” This is an oversimplification of the authors’ results that is more like a newspaper article jargon. What is the definition of “safe” in this context and “promiscuous”? Statements like this should be avoided.

We agree and this phrase is no longer present in the current draft.

Minor issues:

12. Discussion, line 275-276: change “bacteria”  “B. subtilis”. The authors have not shown this for all bacteria, only for B. subtilis.

We agree and this phrase is no longer present in the current draft.

13. Generally, I don't think it is necessary to keep carrying the complete genotype description (amyE::etc) for each strain at each location in the text. This makes it difficult to read. I suggest: define your strain names somewhere and then stick to those names.

We agree and now use Table 2 for technical information on constructs.

14. Results, first paragraph: “genome sequences of the six wild type B. subtilis strains...”. Why is it “the” six strains? Why these strains? Why six? I understand that some of this information is in the methods section, but some explanatory note needs to go into the results section as well to motivate the strain usage.

We have slightly rephrased the introduction of the strains used in this study (see answer to reviewer 1 (L81)). These strains have been isolated from a microscale soil aggregate and therefore are likely to interact in their natural environment. Using six isolates with previously performed kin discrimination typing allowed us to obtain multiple experiments involving both kin and non-kin strain combinations (using more strains obviously would have been better but also significantly more work!).

15. Figure 1a: I presume these are representative images of a particular strain combination. This should be stated in the caption.

The legend of Figure 1 now starts with “a) Representative interactions of control self-interaction (green), kin interaction (blue), and non-kin interaction (red) (ANI indicated)”.

The caption for Figure 1a is too short. Why are there hand-written notes on the petri dishes? Are readers supposed to read them (I can't clearly decipher them)?

Petri dishes were marked for use in the lab. We have now replaced the photos with the petri dishes that were not marked.

16. Figure 1b: please show all individual data points from which the error bars were calculated. Knowing CFU counts, and especially ratios of CFU counts, I am very surprised by the very small error bars. What do the error bars represent?

Error bars represent standard deviation calculated according to three biological replicates, which each constitutes of an average of three technical replicates. Individual datapoints with corresponding average values are indicated in Figs. 1-3.

17. Figure 1b: caption. Please check brackets.

We have completely revised this figure.

18. Figure 2b: please do not abbreviate strain names, to stay consistent with strain names (which are already a bit confusing).

We have completely revised this Figure.

19. Figure 2, 3, caption: the abbreviation KD is never defined. I presume it means kin discrimination.

We no longer use this abbreviation in captions in this revision.

20. Figure 3: please show individual data points. For the PS-196 strain there are no error bars.

We have completely revised all figures and do show individual datapoints in new Figs 1-3. We note some of the error bars in the manuscript are too small to show.

21. Figure 3: Why is PS-13 in the caption but not in the figure itself?

Thanks for spotting this mistake; it is no longer present in the revised figure.

22. Please define ANI on first use.

We now state:

Lines 65-68:

*“...all pairwise combinations of six *B. subtilis* strains isolated from a micro-scale soil population¹⁵ and calculated Average Nucleotide Identity (ANI¹⁶), which is defined as the mean nucleotide identity of orthologous gene pairs shared between two microbial genomes¹⁷ (Table 1).”*

23. Title: Why the review-like title before the “:”? The content of this review-like statement is implicit in the second part of the title. + 24. Title: Something went wrong with the italics in the title.

*We have changed the title to “Kin discrimination promotes horizontal gene transfer between unrelated strains in *Bacillus subtilis*”*

Reviewer #3 (Remarks to the Author):

The authors report that the encounter of non-kin cells stimulates DNA exchanges by transformation in *Bacillus subtilis*. They provide evidence that this occurs through killing of non-kin cells and incorporation of the released DNA. As stated by the authors, killing by non-kin and subsequent import of their DNA has been documented in other species. So, this is not conceptually novel. Yet, it is the first that this is described in *B. subtilis*, suggesting that non-kin predation for the purpose of getting DNA is conserved in naturally transformable bacteria.

*We agree that that our data on coupling of strain lysis and competence in *B. subtilis* is important as it is one of the most important bacterial model systems. However, we would like to add that our finding that the rate of horizontal gene transfer increases with genomic divergence is completely novel. It contradicts the paradigm that recombination is most efficient between more closely related strains (because of sequence homology) and stresses the importance of social interactions regulating competence in determining recombination rate.*

The authors have extensively documented the emergence of recombinants from two non-kin strains at the boundary between the strains swarms. It is interesting that the mechanism of kin-recognition and killing is likely distinct from other documented situations. However, this is mostly left unresolved and the authors are more focused on the evolutionary implications. To support their hypothesis (transformation as a means of genetic mixing), they provide a nice illustration that the recombinants produced at the boundary between non-kin cells could colonize a section of a plate on which the parents strains could not grown. The nature of the experimental system (swarm plates) makes it difficult to address some key points and I am not fully convinced of what is actually going on. Below are some of my concerns:

- The quantification of DNA released at the boundary is technically challenging and I found these experiments inconclusive. The assay in fig 2b technically did not differentiate free DNA released by the dead cells from total DNA contained in the intact cells within the swarm. So, there is no point in comparing the amount of DNA at the boundary and within the swarm.

Samples for DNA measurements were taken from the agar plates and after the agar was melted the cells were removed from the samples first by centrifugation followed by filtration (See Methods lines 370-386). So the DNA measured is from the free DNA and not from the intact cells. We have now removed the swarm DNA measurement data to the supplemental material, as this measurement was performed to obtain the baseline DNA concentration released during swarming.

Indeed, the DNA at the boundary may be short-lived because of cytoplasmic DNAses released by the dead cells. In this respect, the role of “secreted” DNase is also not clear

We show here that after the addition of exogenous DNA to the PS-216:PS-196 interaction, the DNA concentration at the wt pairing is significantly lower than the DNA concentration at the boundary of DNase mutant strains (PS-216Sp Δ yhcR Δ nucB and PS-196Cm Δ yhcR) indicating that extracellular DNAses play an important role in the degradation of DNA (Fig 4b). Of course, we agree, DNAses released from the dead cells could possibly also contribute to the lower DNA concentrations at the boundary.

The amount of DNA at the boundary is compared between two WT and between two mutant strains which both carry mutations for two nucleases (yhcR and nucT, and which are not described in the text).

We added the information to the text, see lines 148-150.

It then cannot be known if the decrease in DNA at the boundary originates from DNase secretion by PS-216 or from the release of DNase by the dead PS-196.

The mutants used in the experiment are PS-216 amyE::p43-cfp, Δ yhcR, Δ nucB and PS-196 sacA::p43-yfp Δ yhcR. The PS-216 carried two mutations and PS-196 one. To answer your question, our data suggests that the DNAses do not come from the PS-196 lysed cells. If they did, we would not see the increase in DNA concentration in the boundary of DNase mutants (as DNA would be degraded by the release of cytoplasmic DNase from lysed PS-196). However, we believe that the source of DNAses is not vital for the conclusions in this experiment.

But most of all, what I found puzzling is that addition of DNA did not affect DNA exchange. This is not consistent with the involvement of natural transformation.

This point was also raised by Reviewer 2. Our findings are consistent with saturating DNA concentrations, where addition of DNA would not make a difference. Saturation was observed previously with a laboratory strain B. subtilis 168 and also with an artificially-enhanced super-competent variant of the 168 strain. For example, transformability of the supercompetent variant of the 168 strain and the parent 168 strain was saturated in the presence of 3000 and cca 2000 ng chromosomal DNA, respectively (Rahmer et al., 2015).

- comGA mutants were used to determine the directionality of the DNA exchange. ComGA has been found to be required for natural transformation but plays additional roles (pilus assembly, persistence, inhibition of cell elongation). The use of a comEC mutant would have been more convincing.

We use comGA mutants in our experiments to deprive one actor of the opportunity to recombine DNA and we believe that comGA mutation in this instance does not affect the growth of the strain (CFU). However, in response to the reviewer's comment, we tested the comEC mutant and we obtained the same results as with the comGA mutant, with similar CFUs and transformation frequencies (see the graph below). Specifically, we constructed comEC mutant (PS-196 sacA::p43-yfp (Cm) ΔcomEC (Ery)) and performed an experiment to test the directionality of DNA transfer between PS-196 sacA::p43-yfp (Cm) ΔcomEC (Ery) and wild type strain PS-216 amyE::p43-cfp (Sp)(Figure R2). The experiment was performed in three biological replicates, each in three technical replicates. Results were compared with previously obtained results for directionality of DNA transfer with comGA mutant (PS-196 sacA::p43-yfp (Cm) ΔcomGA (Ery) and PS-216 amyE::p43-cfp (Sp)). New results using comEC mutant were in accordance with previous results with comGA mutant, which suggest that PS-216 strains relative DNA transfer frequency increases in non-kin combination with PS-196 strain, which suggest that PS-216 strains relative DNA transfer frequency increases in non-kin combination with PS-196 strain when interacting on semisolid B media.

Figure R2: PS-216 transformation frequency when paired with PS-196 comEC and PS-196 comGA mutant strain. (the error bar in comGA experiment is too small to be seen on the figure).

The authors claim that import of DNA from non-kin cells is a consequence of competence induction caused by the non-kin encounter (line 25, 74, 170, 208). I find the evidence supporting this claim rather weak. ComGA is indeed required, supporting a role for natural transformation. However, I do not see evidence for up-regulation of competence genes, or the induction of the competence state. Technically,

the only data available do not show up-regulation of *comGA*, but rather a ~10-fold increase in the number of *ComGA-yfp* positive cells. There is clearly a basal fraction of cells expressing *ComGA-yfp* in the swarm, probably in the competence state. Competent cells of *B. subtilis* have been shown to be tolerant to antimicrobials (PMID: 25899641), the mechanism proposed to be involved in non-kin killing. So an increased number of *ComGA-yfp* may be due to the tolerance of the competent cells to the antimicrobial produced at the boundary, not because *comGA* is induced in a higher number of cells. The regulation of competence in *B. subtilis* is well established and I guess there might be some mutants that could be used to validate/invalidate the induction of competence genes at the boundary.

Thank you for this comment. We have added new data showing the induction of comGA-YFP in one more non-kin and one kin combination (Fig 5d). As for your concern, that the % of PS-216 YFP "ON" cells is an artefact because there are less cells alive in non-kin interaction (and therefore a larger % glow yellow) we checked the raw data on CFU counts of cells carrying the PS-216 comGA-YFP. First, the % of comGA "ON" cells was calculated relative to all PS-216 cells carrying the comGA-YFP reporter (and not total: PS-216 + contact strain). This means that if 1% of cells carrying the comGA-YFP were competent, this subpopulation would be resistant to antibiotics, with the rest of the population carrying comGA-YFP susceptible and it would be reflected in the CFUs of PS-216 comGA-YFP. Second, in all PS-216 interactions, the contact strain shows lower fitness and is killed by PS-216 (Figure 3b). We did however check raw CFU data on PS-216 to refer to the question.

The CFU number of the reporter strain PS-216 comGA-YFP is similar in all interactions (Figure R3a) whether it is paired with a self- strain, a kin strain, or non-kin strain, and this clearly shows that PS-216 is not being killed by the PS-196 strain (Also see figure 3). On the contrary, PS-216 produces an increase in the comGA expression when it meets PS-196 by ~10 fold (at the time we observed the interaction). Also, the number of transformants (Figure R3b) is cca 2log higher in non-kin (red interaction) compared to kin and self-interactions, even though kin and self-interactions include DNA recombination in both interacting strains, whereas in non-kin only the strain PS-216 accepts the DNA (see figure 3). Together, our data suggest that comGA expression is not higher due to lower CFU of the PS-216 strain and that increased comGA activation is not due to lower CFU of the rest of the non-competent population not tolerant to stress.

Figure R3: CFU of transformants compared to CFU of reporter strain PS-216 comGA-YFP at the meeting points with kin, non-kin and isogenic strains. Left: CFU of PS-216 comGA-YFP during kin, non-kin and isogenic self interaction and Right: CFU of transformants obtained during kin, non-kin and isogenic self interaction. All experiments were performed in at least three biological and three technical replicates.

- The link between sensing of membrane damage and induction of competence is not clear. From figure 4a, *sigW* is mostly induced at the boundary by the “prey” strain (PS-196) but transformation occurs mainly in the strain in which *sigW* is not visibly induced, PS-216. How is this explained? Does

secretion of competence pheromones by PS-196 trigger competence in PS-216? If not, please provide direct evidence that sigW induction triggers competence in PS-216.

Thank you for your comment; we agree that we were not clear on the link between competence induction and sigW. We have now additional data, suggesting that Δ sigW mutation in the attacked strain reduces competence in the attacker. This is shown by 1) measuring the comGA-YFP activation of PS-216 paired with PS-196 Δ sigW (Fid 5d), which drops significantly compared to PS-216 paired with wt PS-196 and 2) measuring DNA transfer between PS-216 paired with PS-196 Δ sigW, which similarly to comGA, drops significantly compared to PS-216 paired with wt PS-196 (Fid 5c). We believe this link is now clearer and shows that sigW induction in the attacked strain is important for increased competence and DNA transfer into the attacker. We also added new text in the discussion:

Lines 239-247:

"It is possible that antibiotics that affect cell-envelope integrity induce sigW at the boundary which, together with competence development, increases tolerance to antimicrobials and augment the more competent strain²⁷. It could also be speculated that sigW activated cells with damaged membranes release stress-related metabolites that up-regulate competence in the dominant strain as was documented during inter-species transformation experiments where lysis of E. coli cells and subsequent release of cell metabolites affected the transformation ability of B. subtilis recipient cells²⁸. Similarly, it was recently shown that exposure to antibiotics increases cell-to-cell natural transformation in B. subtilis by affecting the donor strain through as yet unknown mechanism²⁹".

Other comments:

- Student's T test are extensively used all along the manuscript. Unless the authors can show that the data follow a normal distribution, student's T test are inappropriate. Please seek advice from a statistician.

Following the reviewer's advice we took counselling from a statistician. We agree with reviewer that we cannot prove that data follow a normal distribution, however when we processed our data we first calculated an average of three technical replicates (three plates were sampled individually) of each biological replicate (three biological replicates were performed) and afterwards obtained averages: we calculated an average and standard deviation for three biological replicates. Since averages tend to be normally distributed, we believe Student's T-tests are an appropriate statistical method in this case. We also performed a non-parametric Mann-Whitney tests with null hypothesis presuming that transformation frequencies of different strain combinations are the same and alternative hypothesis stating that transformation frequencies of different strain combinations differ from each other (two-tailed test). Exact p-values are given in Table R3. Mann-Whitney tests gave similar results to Student's T-tests shown in the figure and written in the text.

Table R3: p-values for Mann-Whitney tests

	PS-216 (Sp) : PS-196 (Cm)	PS-216 (Sp) : PS-13 (Cm)	PS-196 (Sp) : PS-13 (Cm)	PS-216 (Sp) : PS-218 (Cm)	PS-216 (Sp) : PS-18 (Cm)	PS-216 (Sp) : PS-209 (Cm)	PS-216 (Sp) : PS-68 (Cm)	PS-68 (Sp) : PS-68 (Cm)	PS-18 (Sp) : PS-18 (Cm)	PS-218 (Sp) : PS-218 (Cm)	PS-216 (Sp) : PS-216 (Cm)	PS-196 (Sp) : PS-196 (Cm)	PS-13 (Sp) : PS-13 (Cm)	PS-209 (Sp) : PS-209 (Cm)
PS-216 (Sp) : PS-196 (Cm)	/	/	/	/	/	/	/	/	/	/	/	/	/	/
PS-216 (Sp) : PS-13 (Cm)	0,00004	/	/	/	/	/	/	/	/	/	/	/	/	/
PS-196 (Sp) : PS-13 (Cm)	0,00008	0,00004	/	/	/	/	/	/	/	/	/	/	/	/
PS-216 (Sp) : PS-218 (Cm)	0,0056	0,00004	0,01061	/	/	/	/	/	/	/	/	/	/	/
PS-216 (Sp) : PS-18 (Cm)	0,00004	1,00000	0,00004	0,00004	/	/	/	/	/	/	/	/	/	/
PS-216 (Sp) : PS-209 (Cm)	0,38	0,00004	0,00008	0,007	0,00004	/	/	/	/	/	/	/	/	/
PS-216 (Sp) : PS-68 (Cm)	0,00004	0,11	0,00008	0,00004	0,24	0,00004	/	/	/	/	/	/	/	/
PS-68 (Sp) : PS-68 (Cm)	0,00004	0,38	0,00004	0,00004	0,66	0,00004	0,30	/	/	/	/	/	/	/
PS-18 (Sp) : PS-18 (Cm)	0,00004	0,001	0,00004	0,00004	0,008	0,00004	0,00004	0,00049	/	/	/	/	/	/
PS-218 (Sp) : PS-218 (Cm)	0,00004	1,0000	0,00004	0,00004	0,34	0,00004	0,057	0,24	0,35	/	/	/	/	/
PS-216 (Sp) : PS-216 (Cm)	0,00004	0,48	0,00004	0,00004	0,56	0,00004	0,86	0,66	0,008	0,08593	/	/	/	/
PS-196 (Sp) : PS-196 (Cm)	0,00004	0,05	0,00004	0,00004	0,05	0,00004	0,0001	0,003	0,141	0,70773	0,06450	/	/	/
PS-13 (Sp) : PS-13 (Cm)	0,00004	0,34	0,00004	0,00004	0,43	0,00004	0,73	0,54	0,00008	0,07158	1,00000	0,00399	/	/
PS-209 (Sp) : PS-209 (Cm)	0,024	0,00004	0,00029	0,09	0,00004	0,06253	0,00004	0,00004	0,00004	0,00004	0,00004	0,00004	0,00004	/

- Line 64. Please define KD (kin-discrimination, I suppose)

We have included this definition in the revision.

- Line 82-84. It would just have been easier for the reader to refer to the 3 recognition types previously described (3, 7, 9)

As we have extensively revised the text, this comment is no longer relevant.

- Fig 2b. Please add a legend to the y-axis

We have revised this figure.

- Line 163. The original reference showing that ComGA is required for transformation is Chung and Dubnau, 1998. PMID: 9422590.

Thank you for bringing this to our attention, we have added this reference.

- Line 172. “less-competent strain”. There is no data showing that the strain that generates the fewer recombinant is actually less competent. Have the authors determined the ability of each the two strains to transform using added DNA?

We have removed the phrase from the manuscript.

- Line 270-275. Please consider breaking this long sentence into 2-3 sentences.

This sentence has been removed from the revision.

- Line 288/290. There is no data in this manuscript showing that the tested strains have different phenotypes.

*We now devote second paragraph of the Discussion to phenotypes.
Lines 248-262*

- Line 328. The finding could also be consistent with the “genome-curing hypothesis” (Croucher, Plos Biol, 2016, PMID: 26934590). This hypothesis is never mentioned and should be discussed.

We have mentioned this hypothesis and added this reference to the Discussion, line 273.

References:

- Ansaldi, M. & D. Dubnau, (2004) Diversifying selection at the *Bacillus* Quorum sensing locus and determinants of modification specificity during synthesis of the ComX pheromone. *J Bacteriol* **186**: 15-21.
- Bais, H. P., R. Fall & J. M. Vivanco, (2004) Biocontrol of *Bacillus subtilis* against infection of *Arabidopsis* roots by *Pseudomonas syringae* is facilitated by biofilm formation and surfactin production. *Plant Physiol* **134**: 307-319.
- Cohan, F. M., (2002) Sexual isolation and speciation in bacteria. *Genetica* **116**: 359-370.
- Dubnau, D., (1991) Genetic competence in *Bacillus subtilis*. *Microbiol Rev* **55**: 395-424.
- Gao, S., H. Wu, X. Yu, L. Qian & X. Gao, (2016) Swarming motility plays the major role in migration during tomato root colonization by *Bacillus subtilis* SWR01. *Biological control* **98**: 11-17.
- Goris, J., K. T. Konstantinidis, J. A. Klappenbach, T. Coenye, P. Vandamme & J. M. Tiedje, (2007) DNA-DNA hybridization values and their relationship to whole-genome sequence similarities. *Int J Syst Evol Microbiol* **57**: 81-91.
- Jain, C., R. L. Rodriguez, A. M. Phillippy, K. T. Konstantinidis & S. Aluru, (2018) High throughput ANI analysis of 90K prokaryotic genomes reveals clear species boundaries. *Nat Commun* **9**: 5114.
- Konstantinidis, K. T. & J. M. Tiedje, (2005) Genomic insights that advance the species definition for prokaryotes. *Proc Natl Acad Sci U S A* **102**: 2567-2572.
- Lyons, N. A., B. Kraigher, P. Stefanic, I. Mandic-Mulec & R. Kolter, (2016) A combinatorial kin discrimination system in *Bacillus subtilis*. *Curr Biol* **26**: 733-742.
- Majewski, J., (2001) Sexual isolation in bacteria. *FEMS Microbiol Lett* **199**: 161-169.
- Mirouze, N., Y. Desai, A. Raj & D. Dubnau, (2012) Spo0A~P imposes a temporal gate for the bimodal expression of competence in *Bacillus subtilis*. *PLoS Genet* **8**: e1002586.
- Nijland, R., J. G. Burgess, J. Errington & J. W. Veening, (2010) Transformation of environmental *Bacillus subtilis* isolates by transiently inducing genetic competence. *PLoS One* **5**: e9724.
- Rahmer, R., K. Morabbi Heravi & J. Altenbuchner, (2015) Construction of a Super-Competent *Bacillus subtilis* 168 Using the P mtlA -comKS Inducible Cassette. *Front Microbiol* **6**: 1431.
- Roberts, M. S. & F. M. Cohan, (1993) The effect of DNA sequence divergence on sexual isolation in *Bacillus*. *genetics* **134**: 401-408.
- Solomon, J. M. & A. D. Grossman, (1996) Who's competent and when: regulation of natural genetic competence in bacteria. *Trends in Genetics* **12**: 150-155.

- Stefanic, P., B. Kraigher, N. A. Lyons, R. Kolter & I. Mandic-Mulec, (2015) Kin discrimination between sympatric *Bacillus subtilis* isolates. *Proc Natl Acad Sci U S A* **112**: 14042-14047.
- Stefanic, P. & I. Mandic-Mulec, (2009) Social interactions and distribution of *Bacillus subtilis* phenotypes at microscale. *J Bacteriol* **191**: 1756-1764.
- Tortosa, P. & D. Dubnau, (1999) Competence for transformation: a matter of taste. *Curr. Opin. Microbiol.* **2**: 588-592.
- Zawadzki, P., M. S. Roberts & F. M. Cohan, (1995) The log-linear relationship between sexual isolation and sequence divergence in *Bacillus* transformation is robust. *genetics* **140**: 917-932.

REVIEWERS' COMMENTS

Reviewer #1 (Remarks to the Author):

Line-specific comments:

17: 'surface-dwelling' only in the sense that, like other bacteria, it can't penetrate into most solid substrates, right? See Line 37.

19: Explain here what the 'kin discrimination system' actually does in cases where it succeeds. Cells of one strain kill cells of the other strain, right? It's not reciprocal, right? The killing mechanism does not depend on the pairwise nucleotide identity, but on which strain carries which attack and defense genes, right?

21-23: This sentence is not clear.

66: Here it would be easy to actually describe how the strains were isolated ('from a single 1-cm³ soil sample').

93: Tell the reader what the limit of detection is in each case. This can easily be done by plotting it in the figure as a distinct dashed line for each combination of strains.

106: I don't think there is enough data to support the conclusion that gene flow is primarily unidirectional, from the 'donor' cells being lysed to cells of the lysis-inducing 'recipient'. Only 3 pair combinations were tested for transformation direction. In two of those pairs, 216:196 and 216:218, the lysis-inducing strain 216's TF was raised (3-fold and 15 fold) over its 'self' value of 4×10^{-6} , but the change in TF of the strain being lysed is unknown since we are not told the detection limits. In the third combination, both strains had higher TFs than their self-TF values, a 7-fold increase for strain 13 and a 600-fold increase for strain 196.

111-113: I don't understand this sentence: 'we first compared the fitness of strains (i.e. cell survival - CFU_{strainA}/CFU_{strainA+B}) during nonkin vs self interaction and the fitness of strains in kin vs self- interaction.' Is 'fitness' here measured as CFU_A/CFU_{A+B} (this would seem logical)? Or as 'cell survival' - CFU_A/CFU_{A+B}? If the latter, how is 'cell survival' measured? Is this 'fitness' the same as the 'relative fitness' in Fig. 3b?

114-118: I don't understand how this analysis differs from the analysis described in the next paragraph (lines 119-127) and in Fig. 3b. What do we learn from one that we don't learn from the other?

129: It would be helpful to end this section with a simple summary of who is known to kill who. Briefly, 216 kills 196 and 218, and 13 kills 196, right?

134: The correct abbreviation is DNases, not DNAses.

162-164: These results SUGGEST that DNA concentration might not be limiting, but they're far from conclusive, especially given the uncertainties about how the measurements were performed.

265-6: The cited references did not investigate the recombination effects of the ~1.7% sequence divergence between nonkin in this study. The smallest divergence they looked at was ~3%. Generally, >98% sequence identity is not a hindrance to recombination.

370: More details are needed about the DNA quantification method. It doesn't mention getting rid of the living cells. It also appears to assume that the relevant DNA is contained within the agar, which seems unlikely since most DNA fragments long enough to transform efficiently will not diffuse into the agar after release by cell lysis, but will remain at the cell surface with the cell debris. Any DNA that does diffuse into the agar will not be available for transformation.

Reviewer #2 (Remarks to the Author):

The revised manuscript has changed in many places. One major problem in the original manuscript was the writing, which over-generalized conclusions from the available data, contained lots of instances of inaccurate wording, and used anthropomorphic terminology that was not helpful in my opinion (and that of Reviewer 1). This problem has been fixed in the revised manuscript.

I also think that my request for an explanation for why the increased concentration of extracellular DNA did not affect the transfer frequency (Reviewer 3 had the same comment) has been appropriately addressed by the authors.

The additional experiments presented by the authors to demonstrate transformation and transformation directionality for other non-kin strain combinations (Figure 2), to show how general their findings are, are important additions to the paper. Similarly, the additional experiments for envelope stress and competence induction for the different pairs of strains (Figure 5) make the conclusions of the paper more general.

All my other comments have been appropriately addressed.

In summary, I think this manuscript contains interesting findings for microbiologists and evolutionary biologists. The conclusions are now supported by sufficient data.

Reviewer #3 (Remarks to the Author):

This is a revised manuscript that I initially reviewed in late 2019.

The authors report that the encounter of non-kin cells stimulates HGT by transformation in *Bacillus subtilis*. This is extensively documented by the emergence of recombinants from two non-kin strains at the boundary between the strain swarms. The work shows that the encounter of sufficiently divergent strains of the same species, can engage in antagonistic interactions which stimulates HGT by natural transformation.

Overall, the experiments do support the main conclusions of the manuscript. And the new title now better reflects the content of the manuscript. The manuscript itself is also better written, using more appropriate terminology.

This revised manuscript represents an incremental update, consolidating the previously reported results. More data have been added, using the same methodology as in the original manuscript. More kin and non-kin strain combinations make the results stronger, while keeping the original findings unaltered, but also not providing additional insight.

For instance, the authors further expanded their experimental setup illustrating the production of recombinants at the encounter of non-kin strains. Recombinants are detected as they migrate in a part of the plate in which only recombinants could grow. This is a nice and visual way of documenting the production of recombinants. I did find this experiment essentially illustrative in the first place, and a bit too artificial to draw conclusions about the adaptive value of the phenomenon in ecological situations. Adding more combinations of strains has no added value.

One of my concerns has been partially addressed. The authors conclude that competence is induced in the "dominant" strain upon sensing of the SigW-dependent response of the prey cells. This is a really interesting and key result. It suggests that the dominant strain can sense the damage caused to the non-kin strain and induced competence. This is now further substantiated with data showing that SigW in the prey strain is required for generating recombinants in the dominant strain. The authors see the increase in the recombinants frequency as an indication of the induction of competence. In agreement, they document an increase in the number of comGA-expressing cells, consistent with the idea that at least a sub-population of the dominant strain is

competent. Yet, there is no evidence that this is the result of sensing and activation of the regulatory cascade of competence. In *Bacillus subtilis*, the competence state is controlled by the ComK competence regulator. I felt reasonable to test whether expression of comGA (and/or the increase in recombinants) is dependent of ComK, which would have provided stronger support for the induction of the full competence program through the competence regulatory cascade.

Also I found the following claim of the abstract to be exaggerated, possibly false. "... demonstrating that social interactions can override mechanistic barriers to horizontal gene transfer". It is true that sequence divergence limits recombination, but we are here talking about intraspecific recombination. As far as I know, the little sequence divergence between strains of the same species does not constitute a barrier to HGT. So there is nothing to override here.

Other than that my previously formulated concerns have been addressed.

2nd revision

REVIEWERS' COMMENTS

Reviewer #1 (Remarks to the Author):

Line-specific comments:

17: 'surface-dwelling' only in the sense that, like other bacteria, it can't penetrate into most solid substrates, right? See Line 37.

We used the term surface-dwelling in the sense that it is "terrestrial" or "earthbound" - that its natural habitat is soil.

19: Explain here what the 'kin discrimination system' actually does in cases where it succeeds. Cells of one strain kill cells of the other strain, right? It's not reciprocal, right? The killing mechanism does not depend on the pairwise nucleotide identity, but on which strain carries which attack and defense genes, right?

Right. Different strains produce different toxins which work additively or synergistically and result in largely uni-directional lysis (i.e. of the strain with the least potent cocktail of these effectors). The mechanistic basis of kin discrimination is described in our previous paper, where we show that the presence/absence of these genes correlates with overall genomic divergence (Lyons et al 2020).

We amended the following sentence in the abstract: Genetically distinct *B. subtilis* swarms form a boundary upon encounter, mediated by a fast evolving molecular kin discrimination (KD) system, which is comprised of diverse genes coding for cellular attack and defence mechanisms resulting in killing of the antagonized strain.

21-23: This sentence is not clear

Gene transfer is largely uni-directional and the result of the induction of competence of the recipient strain by the donor strain is linked to a sigW-mediated stress response.

The editor suggested we amend this sentence to:

Gene transfer is largely uni-directional and competence induction in the recipient cell is associated with activation of a stress response in the donor cell, mediated by SigW, a sigma factor known to respond to cell wall stress.

66: Here it would be easy to actually describe how the strains were isolated ('from a single 1-cm³ soil sample').

We changed the sentence to (line 73):

...isolated from a micro-scale soil population...

...isolated from two 1-cm³ soil samples....

93: Tell the reader what the limit of detection is in each case. This can easily be done by plotting it in the figure as a distinct dashed line for each combination of strains.

We have supplied a dashed line in the figure indicating the level of detection.

We added this to the sentence (line 100):

“The transformation frequencies were below the level of detection for some wild type strains (zero transformants were obtained).”

106: I don't think there is enough data to support the conclusion that gene flow is primarily unidirectional, from the 'donor' cells being lysed to cells of the lysis-inducing 'recipient'. Only 3 pair combinations were tested for transformation direction. In two of those pairs, 216:196 and 216:218, the lysis-inducing strain 216's TF was raised (3-fold and 15 fold) over its 'self' value of 4×10^{-6} , but the change in TF of the strain being lysed is unknown since we are not told the detection limits. In the third combination, both strains had higher TFs than their self-TF values, a 7-fold increase for strain 13 and a 600-fold increase for strain 196.

Thank you for the comment. It is true that only 3 pairs were tested (in both directions), but we always measured higher TF in one of the strain-pair. As for the strain being lysed, no TF was reported as no transformants were detected. It is true that probably there might be some transformants in the population of lysing strain, but it is also true that they are being killed (Figure 2b) so gene flow consequently is largely unidirectional (towards the surviving strain). Additionally, we show that even though the strain is lysed (196) it can still be competent above level of detection (when paired with 13), but it is below level of detection when paired with 216. This in our opinion shows that the level of detection is not the limiting factor to determine the unidirectionality of gene flow.

111-113: I don't understand this sentence: 'we first compared the fitness of strains (i.e. cell survival – $CFU_{\text{strain A}}/CFU_{\text{strain A+B}}$) during nonkin vs self interaction and the fitness of strains in kin vs self-interaction.' Is 'fitness' here measured as CFU_A/CFU_{A+B} (this would seem logical)? Or as 'cell survival' - CFU_A/CFU_{A+B} ? If the latter, how is 'cell survival' measured? Is this 'fitness' the same as the 'relative fitness' in Fig. 3b?

We are sorry, but we don't understand what the dilemma is as there is the same equation written for "fitness" and "survival": " CFU_A/CFU_{A+B} " in your question?

Fitness of strain A in interaction with strain B was calculated as: $CFU_A/(CFU_{A+B})$.

First, fitness of strain A was measured in "self-interaction" when strain A was paired with an isogenic version of strain A carrying a different Ab marker (=A'). This is called fitness of strain A in "self-interaction": $fitness_{\text{self}}=CFU_A/(CFU_{A+A'})$.

Next, fitness was measured in a kin interaction, when strain A was paired with a strain B that merged during swarming in the same way: $fitness_{\text{kin}}=CFU_A/(CFU_{A+B})$.

Finally, fitness of strain A was measured during non-kin interaction with a strain B, with which strain A formed a boundary with. $Fitness_{\text{nonkin}}=CFU_A/(CFU_{A+B})$.

Relative fitness:

We wanted to compare how well strain A was doing when paired with a kin strain and how well it was doing when paired with a non-kin strain compared to its performance in isolation. When we compared the fitness values to calculate relative fitness, we divided the fitness calculated for kin with the fitness calculated for the same strain in self interaction (to show there is no significant difference in fitness of strain A when paired with a kin strain). Similarly, we compared the fitness of strain A in non-kin setting to fitness of the same strain in self-setting to show that the strain is either doing better (higher fitness) or worse (lower fitness) in non-kin interaction. To sum up, relative fitness is calculated as $\text{fitness_kin}/\text{fitness_self}$ or $\text{fitness_nonkin}/\text{fitness_self}$. We hope that this explanation is now clearer.

114-118: I don't understand how this analysis differs from the analysis described in the next paragraph (lines 119-127) and in Fig. 3b. What do we learn from one that we don't learn from the other?

Relative fitness (119-127) is just a comparison of the above mentioned values (114-118). We added the fitness values so that the reader would be clearer as how we determined the relative fitness that follows in the second paragraph that was calculated from the fitness values and is shown in the figure 2. Also, when we compare fitness values (119-127) we compare strains between each other, but with relative fitness we compare the same strain in different setting (kin-self, nonkin-self).

129: It would be helpful to end this section with a simple summary of who is known to kill who. Briefly, 216 kills 196 and 218, and 13 kills 196, right?

Thank you for pointing this out, we added the sentence (137-138)

..in non-kin encounters: the PS-216 strain kills the PS-196 and PS-218 strain, and PS-13 kills the PS-196 strain.

134: The correct abbreviation is DNases, not DNAses.

Thank you, we corrected it.

162-164: These results SUGGEST that DNA concentration might not be limiting, but they're far from conclusive, especially given the uncertainties about how the measurements were performed. We have performed experiments in which we add DNA to the agar (3 μ g) and we do not see the increase in TF of otherwise competent strains (transformation is already above level of detection). A similarly experiment was done by Hauser (1994) who showed that saturating levels of DNA are in the area of 0.6 μ g DNA in liquid and if spread on agar for strain *B. subtilis* 168 (Hauser & Karamata, 1994). Another paper reports 2-3 μ g DNA to be saturating and that was used for a strain with induced competence carrying a inducible comK promoter, being 6x more competent than the parent *B. subtilis* 168 strain (Rahmer et al., 2015). This is in line with the fact that *B. subtilis* contains cca 50 sites for DNA uptake (Dubnau *et al.*, 1973, Singh, 1972) and can be in contact with 50 DNA molecules at the same time (Hauser & Karamata, 1994). This, plus the transiency of competence is a clear indication that DNA can be saturating and that cells can't import unlimited numbers of DNA molecules inside the cell.

Also, in DNase mutants, the amount of DNA on the agar is significantly higher, but still TF of otherwise competent strains does not increase (it is already above level of detection btw) (Figure 4b,c,d).

265-6: The cited references did not investigate the recombination effects of the ~1.7% sequence divergence between nonkin in this study. The smallest divergence they looked at was ~3%. Generally, >98% sequence identity is not a hindrance to recombination.

We agree, in Zawadski 1995 the lowest % divergence was 3%, however, in Roberts and Cohan (1993) strains of *B. subtilis* with sequence divergence (sd) between 0,003 and 0,045 were used. In the 168 subgroup (0,003-0.0124 sd) sexual isolation was 0 for the strain 1A2 (receiving its own DNA), 0.02 for strain 1A2 receiving DNA from RO-A-4 (with 0.003 sd) and 0.06 for strain RO-NN-1 (0.0124 sd). RO-E-2 strain with 0.03% sequence divergence showed sexual isolation of 0.19. In the tables below the % sequence divergence at the amyE and sacA locus of strains used in this study are listed. The maximal sequence divergence at the loci above mentioned (amyE and sacA) is 0.024%, which according to the Roberts and Cohan 1993 data is enough sequence divergence for effective sexual isolation.

Sequence divergence at sacA and amyE loci:

sacA	PS-216	PS-13	PS-68	PS-196	PS-218
PS-216					
PS-13	0,000				
PS-68	0,000	0,000			
PS-18	0,000	0,000	0,000		
PS-196	0,017	0,017	0,017	0,017	
PS-218	0,024	0,024	0,024	0,024	0,024
amyE	PS-216	PS-13	PS-68	PS-196	PS-218
PS-216					
PS-13	0,000				
PS-68	0,000	0,000			
PS-18	0,000	0,000	0,000		
PS-196	0,021	0,021	0,021	0,021	
PS-218	0,023	0,023	0,023	0,023	0,011

Additional data on transformation as function of sequence divergence can be found in Majewski and Cohan 1999. Although these experiments were performed with PCR fragments (which can behave differently than genomic DNA with regards to transformation), they also used strains RO-NN-1 and RO-E-1 and compared the frequency of transformation of a wild type 1A96 strain with different donor strains. Within the 1% divergence, frequency of transformation of the wt strain dropped from $1,5 \cdot 10^{-3}$ (self) to $0,8 \cdot 10^{-3}$ and sexual isolation from 1(self) to 1.87. This means that even 1% divergence can be observed and measured as a barrier to DNA integration.

370: More details are needed about the DNA quantification method. It doesn't mention getting rid of the living cells. It also appears to assume that the relevant DNA is contained within the agar, which seems unlikely since most DNA fragments long enough to transform efficiently will not diffuse into the agar after release by cell lysis, but will remain at the cell surface with the cell debris. Any DNA that does diffuse into the agar will not be available for transformation.

The DNA quantification method is described in details in the Materials and Methods (400-401) where we say (see the text in bold to remove the cells):

“Next, they were centrifuged at 10000 g for 5 min and **filtered through a 0.2 µm pore Millipore filter** (Merck, KGaA, Germany) previously wetted with 1 ml of 0.9% NaCl solution.”

We added “to remove the cells” to the text (400-401):

Next, they were centrifuged at 10000 g for 5 min and filtered through a 0.2 µm pore Millipore filter (Merck, KGaA, Germany) previously wetted with 1 ml of 0.9% NaCl solution, **to remove the cells.**

The method used is the most logical: 1) it is the same sampling as we used in other experiments (and we are measuring the transformation frequency at the same time that needs to be sampled in the same way as in other experiments) and 2) the surface of agar is so soft that it would be impossible to scrape the DNA off the 0.7% B media agar plate.

Reviewer #2 (Remarks to the Author):

The revised manuscript has changed in many places. One major problem in the original manuscript was the writing, which over-generalized conclusions from the available data, contained lots of instances of inaccurate wording, and used anthropomorphic terminology that was not helpful in my opinion (and that of Reviewer 1). This problem has been fixed in the revised manuscript.

I also think that my request for an explanation for why the increased concentration of extracellular DNA did not affect the transfer frequency (Reviewer 3 had the same comment) has been appropriately addressed by the authors.

The additional experiments presented by the authors to demonstrate transformation and transformation directionality for other non-kin strain combinations (Figure 2), to show how general their findings are, are important additions to the paper. Similarly, the additional experiments for envelope stress and competence induction for the different pairs of strains (Figure 5) make the conclusions of the paper more general.

All my other comments have been appropriately addressed.

In summary, I think this manuscript contains interesting findings for microbiologists and evolutionary biologists. The conclusions are now supported by sufficient data.

Reviewer #3 (Remarks to the Author):

This is a revised manuscript that I initially reviewed in late 2019.

The authors report that the encounter of non-kin cells stimulates HGT by transformation in *Bacillus subtilis*. This is extensively documented by the emergence of recombinants from two non-kin strains at the boundary between the strain swarms. The work shows that the encounter of sufficiently divergent strains of the same species, can engage in antagonistic interactions which stimulates HGT by natural transformation.

Overall, the experiments do support the main conclusions of the manuscript. And the new title now

better reflects the content of the manuscript. The manuscript itself is also better written, using more appropriate terminology.

This revised manuscript represents an incremental update, consolidating the previously reported results. More data have been added, using the same methodology as in the original manuscript. More kin and non-kin strain combinations make the results stronger, while keeping the original findings unaltered, but also not providing additional insight.

For instance, the authors further expanded their experimental setup illustrating the production of recombinants at the encounter of non-kin strains. Recombinants are detected as they migrate in a part of the plate in which only recombinants could grow. This is a nice and visual way of documenting the production of recombinants. I did find this experiment essentially illustrative in the first place, and a bit too artificial to draw conclusions about the adaptive value of the phenomenon in ecological situations. Adding more combinations of strains has no added value.

One of my concerns has been partially addressed. The authors conclude that competence is induced in the "dominant" strain upon sensing of the SigW-dependent response of the prey cells. This is a really interesting and key result. It suggests that the dominant strain can sense the damage caused to the non-kin strain and induced competence. This is now further substantiated with data showing that SigW in the prey strain is required for generating recombinants in the dominant strain. The authors see the increase in the recombinants frequency as an indication of the induction of competence. In agreement, they document an increase in the number of comGA-expressing cells, consistent with the idea that at least a sub-population of the dominant strain is competent. Yet, there is no evidence that this is the result of sensing and activation of the regulatory cascade of competence. In *Bacillus subtilis*, the competence state is controlled by the ComK competence regulator. I felt reasonable to test whether expression of comGA (and/or the increase in recombinants) is dependent of ComK, which would have provided stronger support for the induction of the full competence program through the competence regulatory cascade.

Thank you for the comment. The *comGA* gene is directly activated by *comK* (Susanna *et al.*, 2004, Hamoen *et al.*, 1998), which directly or indirectly activates more than 100 genes (Berka *et al.*, 2002, Hamoen *et al.*, 2002, Ogura *et al.*, 2002, Hamoen *et al.*, 2003). ComK is essential in competence development, we agree, but without the expression of *comGA* gene, which plays a key role in the binding of transforming DNA to the cell and is required for the construction of the competence associated pseudopilus needed for DNA transport (Briley *et al.*, 2011), transformation does not occur. We believe tracking the gene that actually codes for proteins directly involved in the uptake of the DNA is better than tracking a global transcription factor affecting the activation of more than 100 genes.

Also I found the following claim of the abstract to be exaggerated, possibly false. "... demonstrating that social interactions can override mechanistic barriers to horizontal gene transfer". It is true that sequence divergence limits recombination, but we are here talking about intraspecific recombination. As far as I know, the little sequence divergence between strains of the same species does not constitute a barrier to HGT. So there is nothing to override here.

Please look at our reply to reviewer 1 (265-6).

Other than that my previously formulated concerns have been addressed.

- Ansaldi, M. & D. Dubnau, (2004) Diversifying selection at the *Bacillus* Quorum sensing locus and determinants of modification specificity during synthesis of the ComX pheromone. *J Bacteriol* **186**: 15-21.
- Bais, H. P., R. Fall & J. M. Vivanco, (2004) Biocontrol of *Bacillus subtilis* against infection of *Arabidopsis* roots by *Pseudomonas syringae* is facilitated by biofilm formation and surfactin production. *Plant Physiol* **134**: 307-319.
- Berka, R. M., J. Hahn, M. Albano, I. Draskovic, M. Persuh, X. Cui, A. Sloma, W. Widner & D. Dubnau, (2002) Microarray analysis of the *Bacillus subtilis* K-state: genome-wide expression changes dependent on ComK. *Mol Microbiol* **43**: 1331-1345.
- Briley, K., Jr., A. Dorsey-Oresto, P. Prepiak, M. J. Dias, J. M. Mann & D. Dubnau, (2011) The secretion ATPase ComGA is required for the binding and transport of transforming DNA. *Mol Microbiol* **81**: 818-830.
- Cohan, F. M., (2002) Sexual isolation and speciation in bacteria. *Genetica* **116**: 359-370.
- Dubnau, D., (1991) Genetic competence in *Bacillus subtilis*. *Microbiol Rev* **55**: 395-424.
- Dubnau, D., R. Davidoff-Abelson, B. Scher & C. Cirigliano, (1973) Fate of transforming deoxyribonucleic acid after uptake by competent *Bacillus subtilis*: phenotypic characterization of radiation-sensitive recombination-deficient mutants. *J Bacteriol* **114**: 274-286.
- Gao, S., H. Wu, X. Yu, L. Qian & X. Gao, (2016) Swarming motility plays the major role in migration during tomato root colonization by *Bacillus subtilis* SWR01. *Biological control* **98**: 11-17.
- Goris, J., K. T. Konstantinidis, J. A. Klappenbach, T. Coenye, P. Vandamme & J. M. Tiedje, (2007) DNA-DNA hybridization values and their relationship to whole-genome sequence similarities. *Int J Syst Evol Microbiol* **57**: 81-91.
- Hamoen, L. W., W. K. Smits, A. de Jong, S. Holsappel & O. P. Kuipers, (2002) Improving the predictive value of the competence transcription factor (ComK) binding site in *Bacillus subtilis* using a genomic approach. *Nucleic Acids Res* **30**: 5517-5528.
- Hamoen, L. W., A. F. Van Werkhoven, J. J. Bijlsma, D. Dubnau & G. Venema, (1998) The competence transcription factor of *Bacillus subtilis* recognizes short A/T-rich sequences arranged in a unique, flexible pattern along the DNA helix. *Genes Dev.* **12**: 1539-1550.
- Hamoen, L. W., G. Venema & O. P. Kuipers, (2003) Controlling competence in *Bacillus subtilis*: shared use of regulators. *Microbiology* **149**: 9-17.
- Hauser, P. M. & D. Karamata, (1994) A rapid and simple method for *Bacillus subtilis* transformation on solid media. *Microbiology (Reading)* **140 (Pt 7)**: 1613-1617.
- Jain, C., R. L. Rodriguez, A. M. Phillippy, K. T. Konstantinidis & S. Aluru, (2018) High throughput ANI analysis of 90K prokaryotic genomes reveals clear species boundaries. *Nat Commun* **9**: 5114.
- Konstantinidis, K. T. & J. M. Tiedje, (2005) Genomic insights that advance the species definition for prokaryotes. *Proc Natl Acad Sci U S A* **102**: 2567-2572.
- Lyons, N. A., B. Kraigher, P. Stefanic, I. Mandic-Mulec & R. Kolter, (2016) A combinatorial kin discrimination system in *Bacillus subtilis*. *Curr Biol* **26**: 733-742.
- Majewski, J., (2001) Sexual isolation in bacteria. *FEMS Microbiol Lett* **199**: 161-169.
- Mirouze, N., Y. Desai, A. Raj & D. Dubnau, (2012) Spo0A~P imposes a temporal gate for the bimodal expression of competence in *Bacillus subtilis*. *PLoS Genet* **8**: e1002586.
- Nijland, R., J. G. Burgess, J. Errington & J. W. Veening, (2010) Transformation of environmental *Bacillus subtilis* isolates by transiently inducing genetic competence. *PLoS One* **5**: e9724.

- Ogura, M., H. Yamaguchi, K. Kobayashi, N. Ogasawara, Y. Fujita & T. Tanaka, (2002) Whole-genome analysis of genes regulated by the *Bacillus subtilis* competence transcription factor ComK. *J Bacteriol* **184**: 2344-2351.
- Rahmer, R., K. Morabbi Heravi & J. Altenbuchner, (2015) Construction of a Super-Competent *Bacillus subtilis* 168 Using the P *mtlA* -comKS Inducible Cassette. *Front Microbiol* **6**: 1431.
- Roberts, M. S. & F. M. Cohan, (1993) The effect of DNA sequence divergence on sexual isolation in *Bacillus. genetics* **134**: 401-408.
- Singh, R. N., (1972) Number of deoxyribonucleic acid uptake sites in competent cells of *Bacillus subtilis*. *J Bacteriol* **110**: 266-272.
- Solomon, J. M. & A. D. Grossman, (1996) Who's competent and when: regulation of natural genetic competence in bacteria. *Trends in Genetics* **12**: 150-155.
- Stefanic, P., B. Kraigher, N. A. Lyons, R. Kolter & I. Mandic-Mulec, (2015) Kin discrimination between sympatric *Bacillus subtilis* isolates. *Proc Natl Acad Sci U S A* **112**: 14042-14047.
- Stefanic, P. & I. Mandic-Mulec, (2009) Social interactions and distribution of *Bacillus subtilis* phenotypes at microscale. *J Bacteriol* **191**: 1756-1764.
- Susanna, K. A., A. F. van der Werff, C. D. den Hengst, B. Calles, M. Salas, G. Venema, L. W. Hamoen & O. P. Kuipers, (2004) Mechanism of transcription activation at the *comG* promoter by the competence transcription factor ComK of *Bacillus subtilis*. *J Bacteriol* **186**: 1120-1128.
- Tortosa, P. & D. Dubnau, (1999) Competence for transformation: a matter of taste. *Curr. Opin. Microbiol.* **2**: 588-592.
- Zawadzki, P., M. S. Roberts & F. M. Cohan, (1995) The log-linear relationship between sexual isolation and sequence divergence in *Bacillus* transformation is robust. *genetics* **140**: 917-932.